# Spheroplasts preparation boosts the catalytic potential of a squalene-hopene cyclase

Ana I. Benítez-Mateos[1], Andreas Schneider ®[2], Eimear Hegarty[1], Bernhard Hauer ®[2] ✉ & Francesca Paradisi ®[1] ✉

Squalene-hopene cyclases are a highly valuable and attractive class of membrane-bound enzymes as sustainable biotechnological tools to produce aromas and bioactive compounds at industrial scale. However, their application as whole-cell biocatalysts suffer from the outer cell membrane acting as a diffusion barrier for the highly hydrophobic substrate/product, while the use of purified enzymes leads to dramatic loss of stability. Here we present an unexplored strategy for biocatalysis: the application of squalene-hopene-cyclase spheroplasts. By removing the outer cell membrane, we produce stable and substrate-accessible biocatalysts. These spheroplasts exhibit up to 100-fold higher activity than their whole-cell counterparts for the biotransformations of squalene, geranyl acetone, farnesol, and farnesyl acetone. Their catalytic ability is also higher than the purified enzyme for all high molecular weight terpenes. In addition, we introduce a concept for the carrier-free immobilization of spheroplasts via crosslinking, crosslinked spheroplasts. The crosslinked spheroplasts maintain the same catalytic activity of the spheroplasts, offering additional advantages such as recycling and reuse. These timely solutions contribute not only to harness the catalytic potential of the squalene-hopene cyclases, but also to make biocatalytic processes even greener and more cost-efficient.

Biocatalysis as the use of enzymes to speed-up organic reactions has become a sustainable and efficient alternative to replace or complement traditional chemical catalysis[1]. Typically, biocatalysts can be applied as cell-free enzymes or as whole-cell (mostly bacteria, fungi, yeast) biocatalysts. Cell-free enzymes are preferred to avoid secondary reactions that may happen within the cell resulting in a decrease in the desired product[2]. However, the purification protocols clearly add time and costs to the process. Partial purification of enzymes can also be carried out by simpler methods such as heat shock (for thermophilic proteins) and ammonium sulfate precipitation, often at the expense of the purity degree of the cell-free enzyme[3]. On the other hand, whole-cell biocatalysts are desirable for multi-step transformations that require several enzymes, or for cofactor-dependent

reactions[4]. Likewise, whole cells often protect enzymes from exterior stresses and grant catalytic activity in a more natural environment[5,6].

One of the most challenging biocatalysts to handle are monotopic membrane-bound enzymes[7]. The partial purification of these enzymes and their use as cell-free enzymes is tedious given that the non-soluble enzymes require the addition of detergents and stabilizers to extract the enzymes from the lipid bilayer. Upon solubilization, the additives must still be maintained in the enzyme solution to preserve the structural integrity and activity of the enzyme outside its biological environment. Among the membrane-bound enzymes, squalene-hopene cyclases (SHC) are a class of enzymes with a great potential as biocatalysts to produce high-value flavors, fragrances, and precursors for bioactive molecules[8,9]. The

[1]Department of Chemistry, Biochemistry and Pharmaceutical Sciences, University of Bern, Freiestrasse 3, 3012 Bern, Switzerland. [2]Institute of Biochemistry and Technical Biochemistry, University of Stuttgart, Allmandring 31, 70569 Stuttgart-Vaihingen, Germany. ✉e-mail: bernhard.hauer@itb.uni-stuttgart.de; francesca.paradisi@unibe.ch

biocatalytic potential of SHCs in *Escherichia coli* (*E. coli*) whole-cell environment has been recently reported[10]. Nevertheless, it is well known that the cell membrane acts as a diffusion barrier and sequestering agent for the highly hydrophobic substrates/products, hampering the enzymatic activity[11]. To alleviate the diffusion issues, strategies such as the introduction of transporter enzymes into the cell membrane or the use of additives such as sodium dodecyl sulfate (SDS) are applied to increase cell permeability (Fig. 1). However, such approaches entail time-consuming molecular cloning, tedious downstream steps, and increased process costs[10–12]. Consequently, efficient strategies to circumvent mass transfer issues are of high interest in whole-cell biocatalysis.

Flow biocatalysis is an emerging technology that improves the reaction productivity while minimizing waste and energy consumption[13]. To integrate the biocatalysts into a continuous flow reactor, both cell-free enzymes and whole cells are generally immobilized on a carrier[14]. Since no universal protocol suits the immobilization of every enzyme, diverse strategies have been developed over the last decades that can generally be classified into three main approaches: binding to a premade support, entrapment into a polymer network, and crosslinking[15,16].

A compromise between cell-free enzymes and whole cells is offered by spheroplast preparations: these are gram-negative bacterial cells in which the outer membrane has been partially or completely removed. Despite some applications of spheroplasts reported in biomedicine and cell biology research, their potential role as biocatalysts has been overlooked[17,18].

Here, we present an easy and quick preparation of *E. coli* spheroplasts and their application as biocatalysts to tackle issues of substrate/product diffusion, additives requirement, costly preparation of biocatalysts, and integration into a flow reactor. The performance of the three types of biocatalyst preparation (whole cells, cell-free enzymes, and spheroplasts) is compared under different reaction conditions. As a proof of concept, we employ a membrane-bound enzyme from *Alicyclobacillus acidocaldarius* (*Aac*SHC) for the cyclization of various terpenoids. To optimize the application of this industrially relevant enzyme, different immobilization protocols are tested. As an innovative alternative to traditional immobilization techniques, we introduce the crosslinking of spheroplasts (CLS) as an optimal, cost-effective, and sustainable strategy.

## Results

### *Aac*SHC as a whole-cell biocatalyst in batch and flow

Initially, and based on previous studies on SHC biocatalysis, we defined the cyclization of *E/Z*-geranyl acetone **1E** (trans) /**Z** (cis) into the bicyclic product **2E/Z** as a model reaction to examine the performance of the *Aac*SHC enzyme in the *E. coli* whole-cell environment (Fig. 2)[8,9]. Noteworthy, *Aac*SHC WT presents a much higher preference for the cyclization of the *E*-isomer, thus the maximum conversion that can be expected is 50%[8]. The product formation was analyzed by gas chromatography (GC), revealing only 7% conversion at 10 mM scale after 24 h at 30 °C corresponding to a space-time yield of 0.006 g/L·h$^{-1}$. However, it is a common drawback of SHC biocatalysis that smaller substrate analogs suffer from up to 10$^3$ slower cyclization rates[19].

As our first strategy, we considered continuous processing as a step toward increasing the productivity of the SHC biotransformations[20]. In order to implement the enzymatic reaction into continuous flow, the whole cells were immobilized by entrapment into alginate hydrogel beads. Firstly, the entrapped cells were tested in batch biotransformations reaching the same product formation as the non-entrapped cells (7%) (Supplementary Table 1A). Longer reaction times of 48–72 h were also tested, with no improvement (Supplementary Table 1B). Packing the entrapped cells into a flow reactor and running the cyclization of **1E/Z** in continuous

**a** *E. coli* whole cell biocatalysis

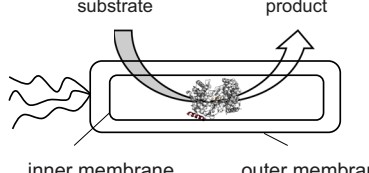

- *High stability*
- Mass transfer limitation
- Low space-time yields

**b** Detergent permeabilization

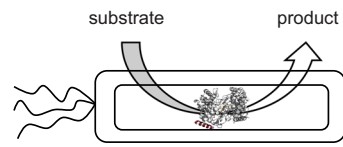

- *Enhanced performance (10-fold)*
- Additional costs
- Additional waste

**c** Cell-free enzyme in membrane-mimic

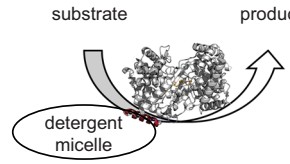

- *Enhanced performance*
- Cumbersome production
- Low stability

**d** This work: Spheroplasts

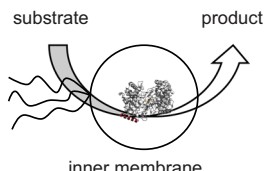

- *Enhanced performance (up to 100-fold)*
- *Simple production*
- *High stability*

**Fig. 1 | Strategies applied in SHC catalysis to avoid mass transfer issues. a** *E. coli* whole-cell biocatalysis with SHC benefits from the high stability of the enzyme in its host. However, mass transfer limitations of the membrane hamper the productivity[43]. **b** Deliberate permeabilization of the cell membrane by using detergents enhances the mass transfer[10] but it may negatively impact the process costs, although further studies are needed to confirm the latter point. **c** The application of isolated enzyme in an artificial membrane mimic enhances the mass transfer but the enzymes suffer from low stability[19]. **d** Spheroplasts comprise a promising alternative to previous applications.

**Fig. 2 | Biotransformation of geranyl acetone 1E/Z with the wildtype AacSHC.** Only the isomer **1E** is converted into product **2E** by the enzyme *Aac*SHC.

mode led however to even lower conversions (Supplementary Table 2). Alternative strategies such as immobilization on agarose and methacrylate did not improve the overall reaction conversion (Supplementary Fig. 1A–C) that was impacted also by the substrate partial affinity for the resin material (in addition to the cell membrane) (Supplementary Fig. 2A, B).

### AacSHC as a cell-free enzyme biocatalyst in batch and flow

To tackle the problem of substrate/product detection when using whole cells, we tested cell-free enzymes as biocatalysts. *Aac*SHC was partially purified according to a previously reported method[21], and applied in batch biotransformations for the cyclization of **1E/Z**. In this case, while the conversion was low (7%) the substrate was fully and reproducibly extracted unlike with whole cells (Supplementary Fig. 3). Then, *Aac*SHC was immobilized on methacrylate microbeads, maintaining up to >99% of its activity when compared to the soluble form (Supplementary Fig. 4). However, yet again the conversion only reached 7%, despite testing several immobilization chemistries and conditions (Supplementary Fig. 5). An additional attempt to perform the biotransformation in flow was done with the best-immobilized system, *Aac*SHC covalently immobilized on methacrylate microbeads (HFA403) (Supplementary Table 3). To facilitate the substrate/product elution from the resin, the flow biotransformation was carried out in a biphasic system (2:1:1 buffer, ethyl acetate, and cyclohexane)[22] without any significant improvement (4% conversion). Triton X-100 has been reported to mimic the natural environment of membrane-associated proteins, as well as increase the water solubility of organic molecules, and it could also prevent the non-specific binding of organic molecules to the resin[23–25]. When this was tested in flow, it did not lead to any improvement. Likewise, increasing the retention time and temperature did not show any significant improvement (8% conversion) (Supplementary Table 3).

### AacSHC spheroplasts: an innovative and more efficient type of biocatalysts

As monotopic enzymes such as the *Aac*SHC are strongly dependent on an intact membrane[26,27], we envisioned a hybrid biocatalyst combining the natural environment of cellular lipidic layer while minimizing the entrapment of the substrate/product in the cell wall. Previous insights on the subcellular location of SHCs indicated that these enzymes are bound to the cytoplasmic membrane when produced in bacteria[21]. Taking advantage of this circumstance, the SHC represented an ideal candidate to explore spheroplasts as a novel type of biocatalysts. The *E. coli* outer membrane could be easily dissolved by simple membrane digestion with lysozyme and ethylene diamine tetraacetic acid (EDTA) that was observed by transmission electron microscopy (TEM) analysis (Fig. 3)[28–30]. Moreover, under the optical microscope, the resulting spheroplasts showed the typical, more circular shape, confirming the loss of the outer membrane (Supplementary Fig. 6A)[28]. Note that about

35–45% of initial whole cells were recovered as spheroplasts, with a significant portion being fully lysed (Supplementary Fig. 7). The presence of active *Aac*SHC within the spheroplasts was confirmed by SDS–PAGE analysis and activity test (Supplementary Fig. 6B). The conversion obtained with the spheroplasts was 1.8-fold higher than using whole-cell biocatalysts, reaching full conversion of the *E*-isomer **1E** to the bicyclic product **2E** at 1–2 mM scale (Table 1). Remarkably, the productivity was 5-fold higher with spheroplast biocatalysts achieving 0.19 g/L of **2E**. Indeed, GC analyses showed that whole-cell biocatalysts retained both substrate and product, while this was completely avoided with the spheroplast preparation (Supplementary Figs. 2B and 8, and GC chromatograms in Supplementary Information).

As the substrate/product extraction was no longer an issue when using spheroplast biocatalysts, we investigated whether the addition of surfactants and other molecules that are typically required to extract the substrate/product from whole-cell biotransformations could be avoided[9]. We found that neither the addition of SDS nor cyclodextrins (CD) influenced the detection of both substrate **1E/Z** and product (Supplementary Table 4) and therefore are no longer essential. Furthermore, the robustness of the spheroplasts under freeze-drying conditions was assessed. After lyophilization and re-hydration, *Aac*SHC spheroplasts showed the same biocatalytic activity as before lyophilization (Supplementary Table 5), offering an excellent storing methodology. In addition, spheroplasts can be stored at 4 °C for at least 2 weeks maintaining the enzymatic activity.

### Spheroplast biocatalysts for the efficient cyclization of terpenes

With these excellent results, we expanded the application of *Aac*SHC spheroplasts as well as three additional SHC variants (*Aac*SHC-G600F, *Aac*SHC-F365C, and AacSHC-A419G Y420C G600A) that were previously developed for the cyclization of geraniol and citronellal[31]. We compared the enzymes' specific turnover frequencies (TOFs) with those of the whole cells, whole cells supplemented with SDS[10], and the cell-free enzyme (Fig. 4). First, the natural reaction of squalene **3** toward hopene **4** was evaluated. Due to its high hydrophobicity, this substrate was barely converted with a TOF of 0.18 h$^{-1}$ with whole cell[8]. Treating the cells with SDS slightly improved the TOF to 0.2 h$^{-1}$. The application of the cell-free enzyme in the presence of the membrane mimic CHAPS improved the reaction 37-fold to 6.6 h$^{-1}$. However, the spheroplasts showed a remarkable improvement of 98-fold in TOF to >17 h$^{-1}$. Next, we examined the promiscuous cyclization of *E,E*-farnesol **5** toward drimenol **6** using the *Aac*SHC-G600F. Whole cells as well as whole cells treated with SDS displayed TOFs of ~2.5 h$^{-1}$. The isolated enzyme and the spheroplasts improved the performance by more than double to 7–8 h$^{-1}$. Pure *E*-geranyl acetone **1E** was converted to **2E** with increasing TOF in the order of whole cells, whole cells treated with SDS, isolated enzyme, and spheroplasts with a maximum of 12-fold improvement to >17 h$^{-1}$. The substrate *E,E*-farnesyl acetone **7** displayed the same tendency with the spheroplasts showing the maximal

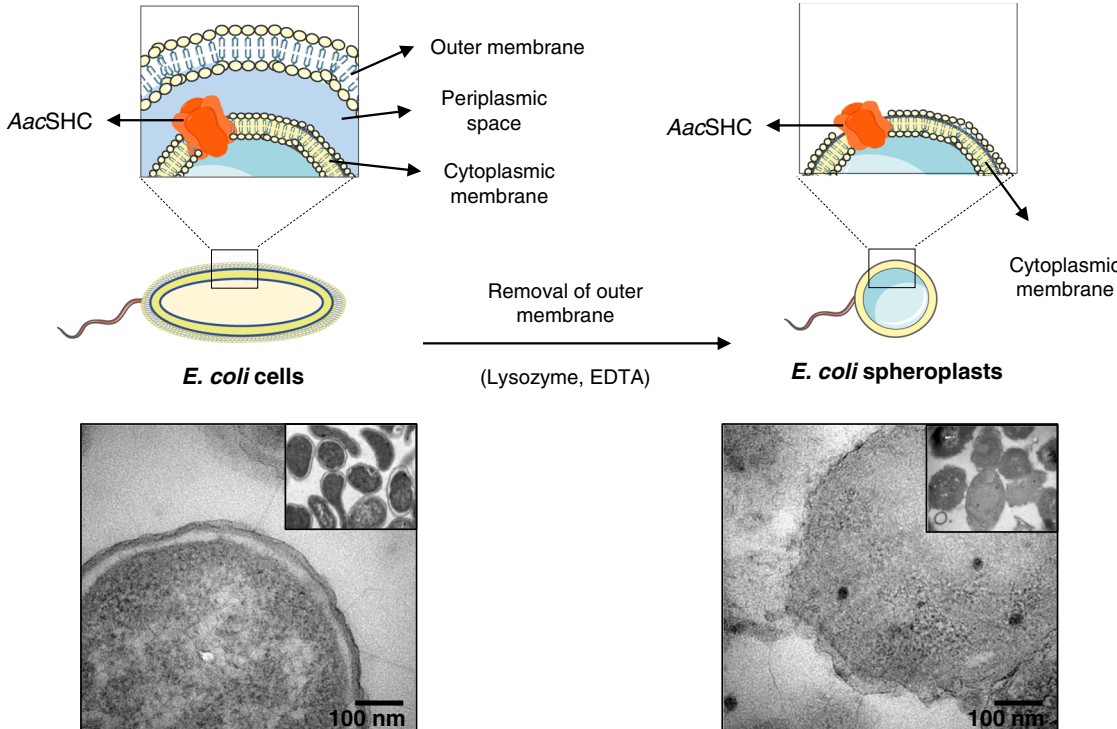

**Fig. 3 | Preparation of spheroplasts by partially removing the outer membrane of _E. coli_ cells.** TEM images show the presence of outer cell membrane on the whole cells (left side) and its almost complete absence on the spheroplasts (right side). TEM images from two independent samples showed similar results. Graphical representations contain modified Servier Medical Art images (smart.servier.com).

improvement of 25-fold in TOF to >17 h⁻¹. In these biotransformations, despite containing a fraction of the biocatalyst, except for compounds **11** and **17**, 40–99% conversion was obtained using *Aac*SHC spheroplasts while this dropped as low as <10% and <30% when using whole cells and free-enzyme biocatalysts, respectively (Supplementary Fig. 9). Next, we tested *E*-geraniol **9** as a smaller substrate analog. In this case, the isolated enzyme *Aac*SHC-F365C was the only preparation to convert this substrate with a TOF of 3 h⁻¹. In contrast, smaller analog (+)-β-pinene **11** underwent cationic rearrangement to (+)-α-pinene **12** with TOFs of 1–2 h⁻¹ using whole cells, whole cells treated with SDS, and free enzyme. Spheroplast preparation improved the TOF 5-fold to 5.5 h⁻¹. We then evaluated the high-value stereoselective cyclization of *E,E*-homofarnesol **13** toward (−)-ambroxide **14**. Spheroplast preparation, with a TOF of 18 h⁻¹, surpassed the performance of whole cells treated with SDS[10] 6.6-fold and the free enzyme 1.4-fold. Prins-En cyclization of (*R*)-citronellol **15** toward (−)-isopulegol **16** catalyzed by *Aac*SHC-A419G Y420C G600A also performed best using the spheroplast preparation with 20.7 h⁻¹ compared to 3.8 h⁻¹ using whole cells and 5.9 h⁻¹ using the free enzyme. Finally, we demonstrated that cyclization of *E,E*-homofarnesoic acid **17** toward sclareolide **18** using spheroplasts is again superior to all other preparations with a TOF of 4.2 h⁻¹ and a maximum improvement of 13.5-fold compared to the whole cells.

To push the reaction further, we increased the substrate concentration to 10 mM. However, congruently to the earlier experiments, the spheroplasts did not show better performance at higher substrate concentrations (Supplementary Fig. 10).

### Entrapped spheroplasts into hydrogel beads
As spheroplasts showed such an excellent performance, we investigated if they could be suited for flow applications. To integrate the spheroplasts into the flow reactor, immobilization by entrapment was applied. Three hydrogel materials (alginate, agarose, and polyacrylamide) were tested in batch. Agarose entrapped spheroplasts showed the best catalytic activity for the cyclization of **1E/Z** into **2E** (45%), followed by the alginate entrapped spheroplasts, while the polyacrylamide entrapped spheroplasts were not successful (Supplementary Fig. 11). In addition, the entrapped spheroplasts were tested in the synthesis of **4** and **8**, the conversion levels for these two cyclizations achieved 40% and 30%, respectively (Supplementary Fig. 10). Hence, the spheroplasts entrapped into alginate beads were selected for the integration into the flow reactor to produce **2E**, but surprisingly no product formation was detected despite different attempted conditions (Supplementary Tables 6 and 7). Nevertheless, the unreacted substrate was observed in the output of the flow reactor.

### A novel technique for enzyme immobilization: crosslinked spheroplasts (CLS)
Inspired by the concept of CLEAS (crosslinked enzyme aggregates) that are carrier-free immobilized enzymes[32], we devised the preparation of CLS. Glutaraldehyde (GA), polyethyleneimine (PEI), and 1,4-butanediol diglycidyl ether (BDE) were employed as crosslinkers to bind the proteins located on the cytoplasmic membrane of different spheroplasts creating a network of spheroplasts (Fig. 5a). No loss of

**Table 1 | Comparison of the efficiency of whole-cell biocatalysts and spheroplast biocatalysts using 1E/Z as substrate**

| Fraction | Protein (mg) | Molar conversion (%) | Biocatalyst productivity in 24 h (×10⁻³ mmol_product/mg_protein) |
|---|---|---|---|
| Whole cells | 0.8 | 28 | 0.7 |
| Spheroplasts | 0.28 | 49 | 3.5 |
| Supernatant | <0.1 | n.d. | n.d. |

Biotransformations with 10 mg of lyophilized biocatalyst (whole cells or spheroplasts, or 10 µL of supernatant containing bacterial periplasm and potential cell debris) in 1 mL of 2 mM **1E/Z**, 1% DMSO, and 20 mM citric acid buffer at pH 6.0. The reactions were incubated at 30 °C for 24 h in glass vials. The amount of protein was determined after extraction and solubilization of *Aac*SHC. Reactions were performed in technical duplicates (SD_spheroplasts: 4%; SD_whole cells: 26%).

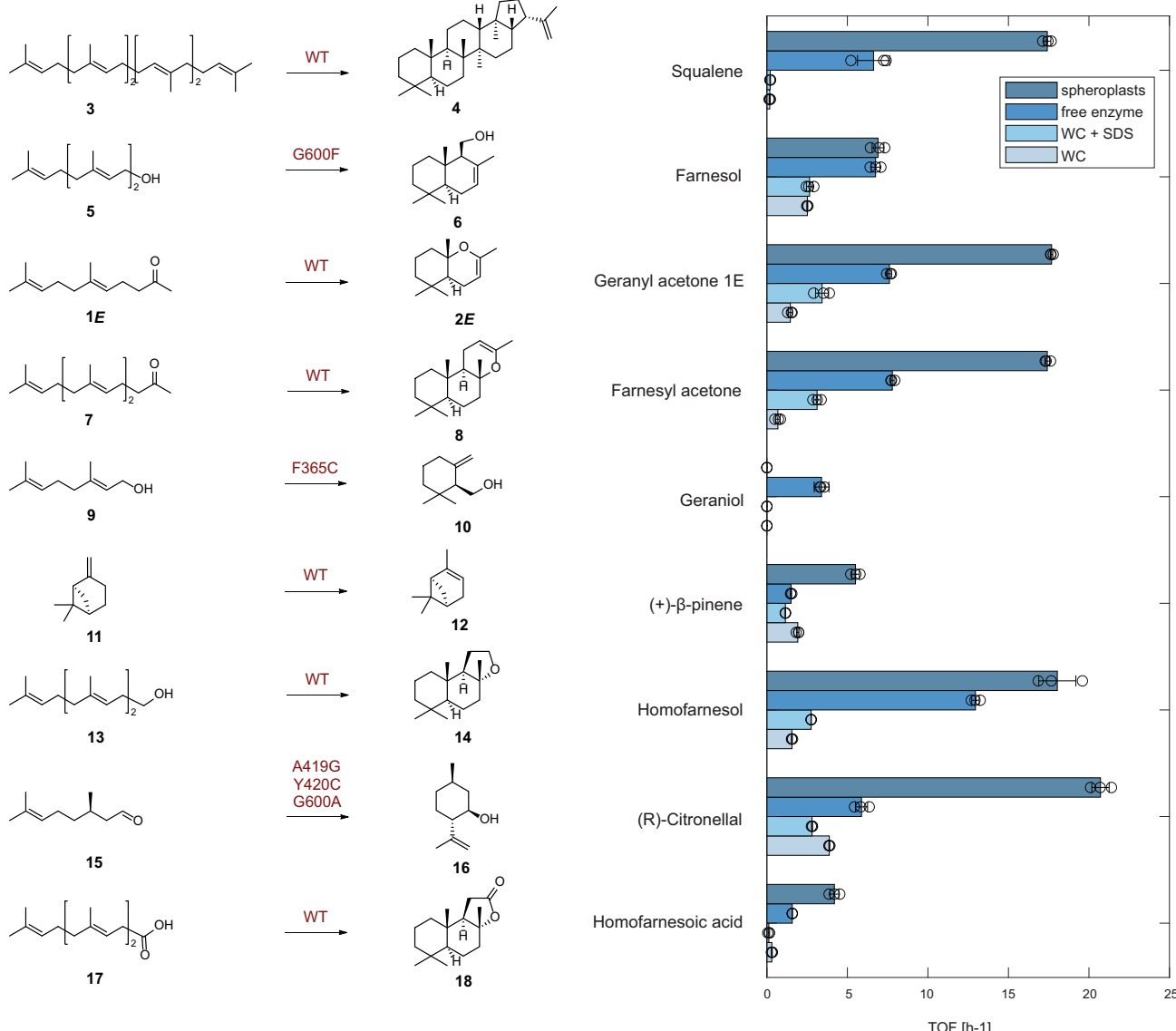

**Fig. 4 | Squalene-hopene-cyclase-catalyzed cyclization of a set of terpenoids.** Squalene **3**, E-farnesol **5**, E-geranyl acetone **1E**, E,E-farnesyl acetone **7**, E-geraniol **9**, (+)-β-pinene **11**, E,E-homofarnesol **13**, (R)-citronellal **15**, and E,E-homofarnesoic acid **17** were used as substrates employing the WT (wildtype) enzyme and the variants G600F, F365C, or A419G/Y420C/G600A as WC (whole cells), WC treated with SDS, cell-free enzyme or spheroplasts. The biocatalyst preparations were compared regarding their TOF (turnover frequency per hour). Reaction conditions: 40 g_CWW/L

whole cells (6 ± 2.7 mg/mL of protein) or spheroplasts (0.7 ± 0.3 mg/mL of protein), 2 mM substrate. The cell-free enzyme (1.3 ± 0.3 mg/mL of protein) was used in 0.2% CHAPS as a membrane mimic. For details see Supplementary Methods. For conversion results, see Supplementary Fig. 9. Reactions were performed in technical triplicates (dot plots). Bars represent mean values ± SD (see SI for more details). *The cell-free enzyme was prepared by the protocol described by Hammer et al.[31]. WT wildtype enzyme, WC whole cells.

activity or efficiency of the spheroplasts was observed following any of the crosslinking protocols. Spheroplasts treated with either GA or PEI showed macroscopic and heterogeneous aggregates (hundreds μm to a few mm) that could be filtered and separated from the bulk, while this was not the case with BDE (Supplementary Fig. 12A–C). Nonetheless, GA crosslinking maintained the spheroplast integrity in the CLS, while this was not the case with PEI where cell debris were observed by TEM analysis (Supplementary Fig. 12D). Yet, both CLS with GA and PEI were applied in biotransformations for the cyclization of **1E/Z** in batch with identical performance as of the original spheroplasts (Fig. 5b). Furthermore, the reusability of the CLS was trialed in consecutive reactions, observing that CLS crosslinked with either GA or PEI were remarkably stable even after washing with buffer and filtration (Fig. 5c).

A clear advantage of CLS is the lack of a solid support. We therefore challenged the requirement of the costly CD that helped

before to solubilize the substrate and prevent binding to the resin material in immobilized preparations. Biotransformations with CLS with or without CD resulted in fact in very similar conversion and CD addition could be avoided altogether (Supplementary Table 8).

## Deciphering the challenge to perform SHC reactions in the flow system

With the efficient spheroplasts biocatalysts and a robust technique for their crosslinking we again attempted to intensify the process to produce **1E/Z** in the flow reactor. In this case, the reaction was performed in absence of additives (i.e., SDS, CD) as we observed before that spheroplast preparations do not require them. Surprisingly, we detected only traces of both the substrate and the product in the output (Supplementary Fig. 13). Considering that the tubing is made of a plastic polymer (PTFE), it was plausible that the substrate may in fact adhere to the surface of the tubing due to its elevated hydrophobicity.

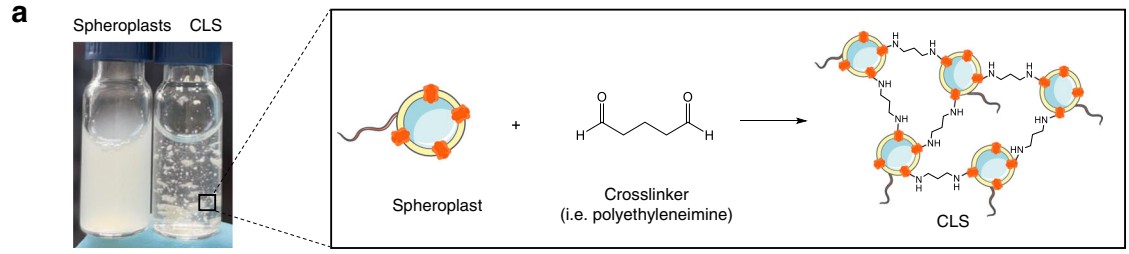

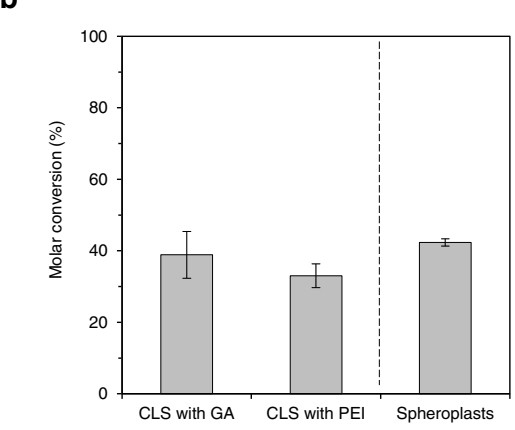

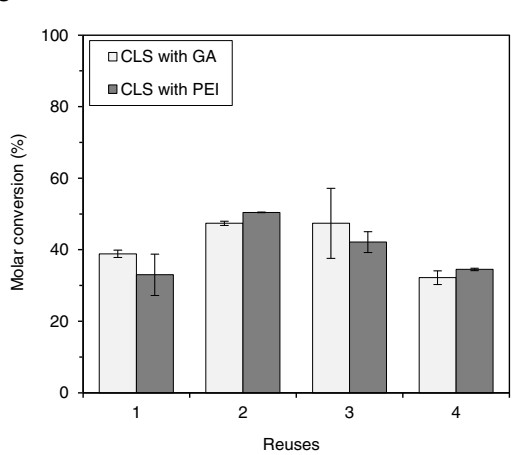

**Fig. 5 | Crosslinked spheroplasts (CLS) as biocatalysts or for the cyclization of geranyl acetone 1E/Z. a** Scheme of the crosslinking of the proteins located on the cytoplasmatic membrane. Glutaraldehyde crosslinking is depicted. Graphical representations contain modified Servier Medical Art images (smart.servier.com). **b** Biotransformations in batch with CLS crosslinked with either GA (glutaraldehyde) or PEI (polyethyleneimine). Reactions were performed in 1 mL of 2 mM geranyl acetone **1E/Z**, 2 mM cyclodextrin, 0.2% SDS and 20 mM citric acid buffer at pH 6.0 after 24 h at 30 °C. The bar chart represents the average of technical duplicates. Data are presented as mean values ± SD. **c** Reuses of the CLS for consecutive biotransformations. Each reuse corresponds to 24 h at 30 °C. After each reuse, the reaction mix was filtered and replaced by a fresh substrate solution. Spheroplasts could not be reused due to the lack of crosslinking; thus, the biocatalysts were lost in the flow-through after the first filtration. The bar chart represents the average of technical duplicates. Data are presented as mean values ± SD.

In fact, as an incise, it is important to mention that all batch reactions must be carried out in glass vials rather than plastic vessels for the same reason. To prove such hypothesis, the flow reactor was fed with a substrate solution in the absence of additives, in presence of CD, and in presence of SDS. Samples were taken at different points along the tubing of the flow system before reaching the reactor (Supplementary Fig. 14A). We observed that only the addition of SDS prevented to some extent the adhesion of the substrate **1E/Z** to the plastic of the tubing (Supplementary Fig. 14B). However, the substrate detected was consistently decreasing along the tubing regardless of the presence of any additive and, with our flow system set-up this reaction could simply not be implemented.

## Discussion

To overcome the drawbacks of the outer cell membrane while maintaining SHC in its membrane environment, we have found a "goldi-locks" compromise between the use of cell-free enzymes and the use of whole-cell biocatalysts: the spheroplasts. Although the preparation of spheroplasts may be further enhanced to reduce complete cell lysis, this proof-of-concept work showcases the potential of spheroplast preparations for enzymatic catalysis. We have adapted a known solution for a novel application as efficient biocatalysts for the cyclization of geranyl acetone **1E/Z** into **2E**, reaching up to 1.8-fold higher specific activity with 2.9 times less amount of biocatalyst (Table 1). The straightforward and quick protocol to prepare spheroplasts makes this strategy highly attractive to expand its application scope to other membrane-bound enzymes and for its implementation in industrial processes. This is particularly relevant for SHC reactions which are getting great attention in the flavors industry to synthesize

enantiopure cyclic terpenoids but have remained a challenging catalyst until now[33]. Moreover, despite the added cost of the spheroplasts and CLS preparations versus whole cells, this approach appears to be a more cost-efficient and sustainable alternative due to the simpler reaction set-up and the reusability of the CLS (Supplementary Tables 8 and 9). Besides, their proven stability after lyophilization makes them storable and their application is as simple as batch chemistry.

The potential of the SHC spheroplast biocatalysts was also proved with eight additional terpenoid substrates, including the high-value cyclization of E,E-homofarnesol **13** to produce the odorant (−)-Ambroxide **14**[10] that provided valuable information in a comparative evaluation of four biocatalyst preparations (whole cells, whole cells + SDS, cell-free enzymes, and spheroplasts). First, the treatment with SDS did not result in better TOFs compared to the whole cells using the substrates squalene **3**, E-farnesol **5**, (+)-β-pinene **11**, (R)-citronellal **15** and E,E-homofarnesoic acid **17** which discloses a limitation of the membrane permeabilization using detergents (Fig. 4). Indeed, Eichhorn et al. demonstrated that among various tested detergents, SDS was the only one that permitted the cyclization of homofarnesol **13** toward (−)-Ambroxide **14**[10]. Second, using cell-free enzymes with different substrates and different variants, always resulted in superior performance compared to the whole-cell preparation. (+)-β-pinene **11** represents an exception in this regard, which may be accounted to its small and hydrophobic structure and the entailed diffusion behavior through micelles[34]. A screening of alternative detergents by high-throughput methods such as described by Kotov et al.[35] could improve the diffusion limitation in vitro. Interestingly, E-geraniol **9** was only converted using the cell-free enzyme that was prepared using a different protocol[31]. This fact highlights the importance of enzyme

preparation in membrane-bound enzyme catalysis. Finally, the natural substrate **3** was very poorly converted using SHC whole cells, due to the known challenge of substrate diffusion through the outer membrane[36]. Noteworthy, the spheroplasts improved the reaction in almost every case by up to ~100-fold, which highlights this superior hybrid enzyme preparation. The only exception was the cyclization of **9**, which was exclusively transformed by tailor-made cell-free enzymes.

Regarding the immobilization of SHCs, no previous studies have been reported to date. Herein, we performed an extensive analysis of different immobilization strategies, from entrapment into hydrogels to attachment to a solid support (Supplementary Table 10). Whereas the hydrogel entrapment showed good results to keep the enzymatic activity after immobilization, this strategy failed the stability test (Supplementary Fig. 11). More robust biocatalysts have been developed by attachment to solid supports (methacrylate and agarose microbeads). However, we found a strong unfavorable affinity of the substrate/product to the support. This issue could be alleviated by using more hydrophilic materials such as agarose, but still some 'sticking' effect happened. In some of our previous works, we have observed a similar phenomenon when using non-polar substrates that can interact with hydrophobic supports[37,38]. Therefore, we can conclude that SHC-catalyzed reactions are not particularly compatible with carriers used to immobilize the biocatalyst. As an innovative alternative, we developed the concept of CLS, which overcomes the stability and reusability issues of the catalysts, it avoids any interaction between the substrates/products and the carrier, and of course, also eliminates costs and waste management linked to the use of a carrier (Fig. 5).

Overall, the strategy to enhance the catalytic activity of the SHC by implementing it in flow mode did not result in satisfying findings. Surprisingly, longer residence times did not improve the overall conversion in any case, which could suggest strong inhibitory effects on the SHC as devised by Neumann and Simon[19]. However, in our opinion, the major challenge of these specific reactions is the hydrophobicity of the substrates/products which readily diffuse in the *E. coli* cell membrane[34,39]. In this regard, segmented flow techniques as presented by Tang et al. may offer one solution[40]. However, that study was based on hydrophilic terpenes and a soluble class I cyclase, while *Aac*SHC is strongly dependent on the membrane structure. More difficult to overcome are the inherent limitations of the plastic tubing material of many flow systems which also sequester the substrate/product, even when the catalyst could be prepared as CLS. Despite the limited commercial availability of tubing material for flow-reactor systems, further investigations into alternative tubing material will be done.

In summary, we bring an innovative approach to develop more efficient and sustainable biocatalysts by removing the outer layer of gram-negative bacteria. During this research journey, we have also collected valuable insights into the optimal operation conditions of membrane-bound enzymes, and their limitations, specifically about SHCs. Thus, the notoriously challenging stereoselective head-to-tail cyclization could be finally added to the chemical toolbox and release terpene synthesis from the classical ex-chiral pool approach[41,42]. Furthermore, spheroplasts are not limited to membrane-bound enzymes: they can be a potential solution for other relevant biotransformations mediated by cytoplasmic enzymes in whole-cell systems that suffer from the drawbacks of the outer cell membrane barrier. Finally, we have introduced an immobilization strategy for spheroplasts (CLS) with potential application to any other biocatalyst.

## Methods

### Materials
All the reagents used for syntheses, buffer preparation, culture media preparation, and biochemical work were purchased from Carl-Roth (Karlsruhe, DE), VWR (Pennsylvania, US), Sigma-Aldrich (St. Louis, US) and Alfa-Aesar (Ward Hill, US). The substrate (E/Z)-geranyl acetone was obtained from Combi-blocks (San Diego, USA). All the other substrates were chemically synthesized and analyzed by $^1$H-NMR, $^{13}$C-NMR, and GC/MS.

### Protein production[9]
The plasmid pET22b(+) harboring the gene of *Aac*SHC (UniProt: P33247) or a variant was transformed into *E. coli* BL21(DE3) by heat shock at 42 °C for 45 s followed by ice cooling for 3 min. Individual colonies were picked from generated agar plates and cultivated in 10 mL LB medium with 100 µg/mL Ampicillin overnight at 37 °C, 150 rpm. Then, 1 L flasks containing 300 mL of T-DAB autoinduction medium with lactose as the inductor and 100 µg/mL Ampicillin were inoculated with 3 mL of the overnight culture. The cultures were incubated for 20 h at 37 °C, 150 rpm, and harvested afterward (4000 g, 20 min).

### Enzyme purification by thermolysis[3,9]
The cells were resuspended in 1 mL of Lysis buffer (200 mM citric acid, 0.1% EDTA, pH 6.0) and incubated for 1 h at 70 °C. The cell suspension was centrifuged (14,000 g, 1 min) and the supernatant was discarded. As the enzyme is membrane-bound, 1 mL of 1% CHAPS buffer was added to extract it from the cell pellet by shaking at room temperature for at least 1 day at 600 rpm. After centrifugation (14,000 g, 1 min), the supernatant containing the *Aac*SHC was transferred to a new tube followed by SDS−PAGE analysis and determination of enzyme concentration by using the EPOCH2 (nanodrop Tek3 plate). Therefore the "Protein A280" mode was chosen with MW = 71,439 Da and molar extinction coefficient ε = 185,180 as protein-specific data.

### Preparation of spheroplasts
Based on previous protocols[28,29], 100 mg harvested or lyophilized cells were resuspended in 1 mL of 20 mM citric acid at pH 6.0 with 10% sucrose and 150 mM NaCl. After centrifugation (15,000 g, 3 min), the cells were resuspended in 20 mM citric acid at pH 6.0 with 10% sucrose, 1 mM EDTA, and 1 mg/mL of lysozyme. The suspension was incubated for 30 min at room temperature followed by centrifugation for (15,000 g, 5 min). The supernatant containing the outer membrane was discarded. Finally, the resulting spheroplasts were washed (3×) with 2 mL of 20 mM citric acid at pH 6.0.

The change of the cell shape corresponding to the removal of the outer membrane was confirmed by TEM (Fig. 3) and optical microscopy (Supplementary Fig. 6A). *E. coli* cells (rod shape) and the spheroplasts (circular shape) were visualized using transmission light in a Nikon Ti2 Eclipse microscope with the objective 60× (oil).

### SDS−PAGE
After protein purification and extraction, 20 µL of the enzyme/cell solution was mixed with 20 µL SDS loading buffer and heated to 95 °C for 5 min. Afterward, 5–15 µL of the preparation was loaded on the 12% SDS−PAGE.

### Biotransformations in batch mode
In all, 1 mL of the reaction mix containing the substrate (from a stock solution in DMSO) and citric acid buffer at pH 6.0 was added to a glass vial. The reaction mix could also contain SDS, CD, or triton as specified for each experiment. Typically, 10 mg of whole/cells or spheroplasts were added to the reaction, unless otherwise specified. The biotransformations were incubated at 30 °C for 24 h with shaking, unless otherwise specified. Bar charts and dot plots in Fig. 4 and Supplementary Fig. 9 were plotted using Matlab R2022b (9.13.0.2049777). SDs were calculated in Excel (Microsoft Office 18).

## Preparation of crosslinked spheroplasts

In all, 100 mg of lyophilized spheroplasts were added in 2 mL of 100 mM citric buffer at pH 6.0 containing the crosslinker (PEI 60 kDa 50% aq. sol.: 100 mg/mL; BDE: 0.5 M; GA: 1%). The reaction was incubated at room temperature for 5 h (16 h for BDE) under shaking, and after centrifugation (15,000 g, 2 min), the supernatant was discarded. The resulting CLS were washed (5×) with 2 mL of 20 mM citric buffer at pH 6.0. See Supplementary Fig. 12 for more details on the CLS size.

## Biotransformation in continuous flow mode

Flow reactions were performed using an R2S/R4 Vapourtec flow reactor equipped with a V3 pump and an Omnifit glass column (6.6 mm i.d. × 100 mm length) filled with the immobilized biocatalyst (1–2 g) as a packed-bed reactor (PBR). To fill the PBR with the desired immobilized biocatalyst, a suspension was made in 20 mM citric buffer at pH 6.0 and then the suspension was transferred to the reactor by pipetting. A first equilibration step was performed by running 20 mM citric acid buffer pH 6.0 buffer at 0.5 mL/min for 10 min. Then, the solutions of substrates at different concentrations were mixed in a T-tube and pumped through the PBR containing the immobilized biocatalyst. The flow rate was adapted depending on the desired residence time for each reaction. Samples were collected after each column volume and analyzed by GC.

## Gas chromatography (GC)

Samples were extracted with ethyl acetate:cyclohexane (1:1) in a final volume of 1 mL and the resulting organic phase was submitted to GC analysis. Agilent GC8860 equipment was employed for the analyses, with an Agilent19091J-413 column (30 m × 320 µm × 0.25 µm) and nitrogen as carrier gas (pressure: 12.816 psi), unless otherwise specified. Injections (1 µL) were performed in split mode (5:1). The following temperature profile was used: 1 min at 155 °C, 11 °C/min to 205 °C, 0.6 min at 205 °C; inlet and detector temperature: 250 °C.

The conversions (%) were calculated directly from GC spectra (Supplementary Figs. 15–24) by integration-quotient of substrates and products. The molar conversions were also calculated by using standard curves of the substrates and products (1–15 mM) when possible.

The results presented in Fig. 4 and Supplementary Fig. 9 were obtained by GC analysis with an Agilent 7820A equipped with a mass spectrometer MSD5977B and a HP-5MS capillary column (Agilent, 30 m × 250 µm × 0.25 µm) and helium as carrier gas with a constant pressure of 14.168 ψ. Injections (1 µL) were performed in split mode (10:1).

## Reporting summary

Further information on research design is available in the Nature Research Reporting Summary linked to this article.

## Data availability

All data are available in the article and its Supplementary Information file; data are also available from the corresponding authors upon request. The squalene-hopene cyclase protein data used in this study are available in the Uniprot database under accession code P33247. Source Data (scan of uncropped gels) are provided with this paper.

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

## Acknowledgements

The authors are grateful to the SNSF (200021_192274, F.P.) and the University of Bern ("Seal of Excellence Fund" Postdoctoral Fellowship SELF19-03, A.I.B.-M.). We gratefully acknowledge the Deutsche Forschungsgemeinschaft (DFG HA 1251/6-1, B.H. and A.S.) for research funding. We thank Prof. Eva Hevia and Prof. Christoph von Ballmoos for providing equipment for GC analysis and optical microscopy, respectively. Electron microscopy sample preparation and imaging were performed with devices supported by the Microscopy Imaging Center (MIC) of the University of Bern.

## Author contributions

F.P. and B.H. conceived and guided the project. A.I.B.-M. performed most of the experiments and drafted the manuscript. A.S. and E.H. contributed to the experimental work. A.S. contributed to the draft writing. All authors discussed and agreed to the final version of the manuscript.

## Competing interests

The authors declare no competing interests.
