## [Peer Review File · Nature Communications]

Spheroplasts preparation boosts the catalytic potential of a squalene-hopene cyclaseREVIEWER COMMENTS

Reviewer #1 (Remarks to the Author):

Spheroplasts preparation boosts the catalytic potential of a terpene cyclase
Ana I. Benítez-Mateos, Andreas Schneider, Eimear Hegarty, Bernhard Hauer*, and Francesca Paradisi*

The authors address the use of spheroplasts in squalene hopene cyclase-catalysed reactions. These enzymes have attracted the attention of chemists and biochemists for several decades because unlike other terpene cyclases they do not require phosphorylated precursors, but proceed via a Brønsted acid catalysis-driven mechanism. Many reports are available showing the plasticity of these enzymes and their ability to be evolved through enzyme engineering resulting into tailor-made biocatalysts with modified substrate specificity and/or product selectivity. A recent example showed as well that these enzymes can be evolved to biocatalysts usable under process-relevant conditions in the field of flavours and fragrances.

- What are the noteworthy results?

SHC enzymes are associated to the inner membrane of bacterial membranes e.g. when produced in *E. coli* for the use in biocatalytic reactions: the substrate has to access the enzyme through both outer and inner membranes. Cell permeabilization is a technique that is often addressed in the field of biocatalysis, whereas as the authors do mention carrying out biocatalytic reactions with spheroplasts is generally not considered. The authors demonstrate the advantage of using spheroplasts over whole cells (*E. coli* cells producing an SHC enzyme): the use of spheroplasts allows for a significant increase in activity. The authors introduce also the new concept of cross-linked spheroplasts and spheroplasts immobilization for the use in flow catalysis, flow reactions being of increasing interest in the field of biocatalysis.

- Will the work be of significance to the field and related fields? How does it compare to the established literature?

The advantage of using of spheroplasts over whole cells reported in the present study may trigger an increase in interest for this biocatalyst formulation in biocatalytic reactions/processes in general. Novelty is addressed: reports on use of spheroplasts in biocatalytic reactions are rare/inexistent.

At this early stage of investigations is perhaps less evident the link between the use of spheroplasts and biocatalysis in flow. As mentioned by the authors more investigations must be done in this direction to confirm their applicability when immobilized and in flow reactions. Using SHC enzymes as target for such investigations is a difficult task due not only to the SHC enzyme itself, which is membrane-associated and handling water-insoluble terpenoid substrates, but also due to the currently used flow catalysis setup: reactivity/adsorption of terpenoids with plastic materials.

- Does the work support the conclusions and claims, or is additional evidence needed?

The work supports the conclusions and claims regarding the better activity when considering SHC catalysis in the context of spheroplasts. It surely would have been interesting in this regard to extend/diversify the range of substrates used outside those tested and listed in Figure 2, which appear to be limited to squalene (AacSHC prototype substrate), C10 and C15 terpene-alcohols and -ketones. Using substrates as listed e.g. in Syren et al (2016), *Current Opin. Struct. Biol.* 41, 73-82 would have strengthened further the view of a broad applicability of the methodology, extending also the functional groups of substrates to aldehydes, acids, or epoxydes with epoxygeraniol, or aromatic substrates, cyclic compounds such as pinene, industrially-relevant substrates like homofarnesic acid, homofarnesol, and citronellal, substrates for the synthesis of sclarelolide, (-)-Ambrox and menthol,

- Are there any flaws in the data analysis, interpretation and conclusions? Do these prohibit publication or require revision?

Some points remain open, which after clarification/revision would turn this good into an excellent piece of work (details below in "additional remarks").

- Is the methodology sound? Does the work meet the expected standards in your field?

Yes regarding the use of spheroplasts applied to SHC catalysis for boosting activity. Regarding biocatalysis in flow, it is not surprising that the use of (certain) plastic materials is an issue when working with terpenes/terpenoids in general. But the authors recognized the limit of the system and the urgency of adapting the usually used flow systems for working with terpenoids. This is an incentive to evolve further the design of flow biocatalysis setups.

- Is there enough detail provided in the methods for the work to be reproduced?

Yes

Shall the comments I have below be addressed by the authors, I think this manuscript would be suited for publication in Nature Communications.

Comments to authors:

Manuscript

Title:

It is possibly preferable to name the enzyme the authors work with directly in the title, i.e. "squalene hopene cyclase" and not using the general term "terpene cyclase". Other terpene cyclases are not addressed in the submitted work. The question is if spheroplasts are relevant to working with terpene cyclases other than SHCs or oxidosqualene cyclases as these enzymes will use activated compounds as their substrates, which for my understanding are produced within the cells as part of their metabolism in biosynthesis processes, correct?

Page 2 lines 38-42

The statement suggests an obligation/advantage of purifying membrane-associated/membrane-bound enzyme for efficient use in biocatalytic reactions, which is a difficult and costly task, questioning as mentioned enzyme stability when purification. But: is purification of such enzymes indeed required? Membrane-associated enzymes can be considered as "immobilized" per se. If using whole cells is an issue due to side reactions, then working with a membrane fraction may possibly solve this problem. There is to my understanding no need to justify with this statement the present work, which introduces the novel concept of using spheroplasts in biocatalytic reactions for boosting activity.

Page 2 lines 46-49

Cell membrane/diffusion barrier. If it is well understood that transporter enzymes would help for the transport of water-soluble compounds through the lipophilic cell membrane, would such transporters indeed allow for a better access of lipophilic substrates to the inner-membrane-associated SHC enzyme? These substrates would certainly well dis-solve/diffuse into the cell membrane. Would it be better for this reason to state here that lipid bilayers act rather as a sequestering agent for these compounds (as mentioned later e.g. Page 4, lines 103-107) rather than speaking of a diffusion barrier for lipophilic compounds?

Page 3, Scheme 1. The representation of the mixture of geranylacetone isomers may not be the best (same later on in the manuscript and supplementary information). Should the two isomers better be shown for clarity? Nomenclature: why introducing t/c (cis/trans) as the manuscript mentions often the E-isomer (also in legend to scheme 1). Why not better stick for consistency/clarity to this E/Z nomenclature throughout the manuscript and not use t/c?

Page 3, lines 93-95. Is it possible that entrapment adds an additional barrier to the diffusion of substrates to the enzyme?

Page 4, lines 112-115. Flow biotransformations are carried out in a mixture of buffer and solvents. Did the authors test the influence of solvent addition on SHC activity, e.g. if the solvents used inhibit activity?

Page 4, 2nd paragraph. Do the % conversion values reported really allow to include wording like "activity improvement" or "effect on conversion" as all the values are between 4 and 7%, probably

being within the error margin inherent to the system?

Page 4, line 124. Are proteins expressed in bacteria? This terming became common language, but isn't it of general understanding that genes are expressed resulting into protein/enzyme production.

Page 4, line 129/131. From the text it is not straight forward how biotransformation yield and productivity are related. Could biotransformation yield here better be named conversion as it is in Table 1?

Page 4, line 133. Did whole cells retain substrate and product indeed, or could have an alternative extraction method allow for a better extraction?

Page 5, first paragraph. Spheroplasts storage: how well are spheroplasts storable at 4°C?

Page 5, Table 1. Why not indicating for clarity in the Table column header "molar conversion (%)" in full instead of using a not trivial abbreviation (m.c.)? This applies throughout Tables/Figures in the manuscript and supplementary information (wherever the abbreviation is not spelled out). Productivity: are the reported values not micromol product/mg enzyme: change unit in table header for straightforward reading? Specific activity: calculated on 24h. Isn't it better to calculate this value over the linear reaction rate as possibly a plateau might be reached at different timepoints depending on the biocatalyst formulation considered? How do "productivity (24 h)" and "specific activity calculated in 24 h biotransformations" correlate? This must be clarified.

Page 6. It would have been relevant to extend the range of examples of substrates (size, functional groups) to generalize the concept of "spheroplasts as biocatalysts for the efficient cyclization of terpenes".

Page 7, lines 182-184. Spheroplasts use at higher cell concentration. It looks like increasing substrate concentration has a negative effect on catalytic activity (Figure S8), the relative conversion decreases with increasing substrate concentration. Is this correct, and is there an explanation for this? Could this be a general rule? Was this expected? Here the use of whole cells seems to be of an advantage. Do the reported values in the text really match the ones displayed (conversion) in Figure S8: better conversion with agarose entrapment for cyclization of 3 to 4, slightly better conversion with alginate entrapment for cyclization of 7 to 8? Why was then alginate entrapment used for entrapment for cyclization of 1 as the result suggest that best method for entrapment could depend on substrate considered? Is this conclusion correct?

Page 7, spheroplasts entrapment in hydrogel beads. Does entrapment add a new barrier to substrate transfer to the enzyme?

Page 8, SHC reactions in flow system. Reactivity of plastic materials with terpenoids is known and requires some precautions.

Page 9, line 246. Is it the case that ref 33 states that SHC reactions have remained a challenging task until now? Isn't it rather the case that these enzymes were simply to date mainly studied for their plasticity/evolvability rather than for their applicability at large scale (process-suitability)?

Page 9, lines 246-249. True: no additives added to the biotransformation itself. But the preparation of spheroplasts requires additional preparation compared to whole cells, involving sucrose, NaCl, EDTA and lysozyme. Is it therefore possible to directly conclude in general on the more cost-efficient and sustainable alternative of using spheroplasts vs. whole cells? This can, but must not, be the case depending on the required "additives" (Table S8 which is referred to only mentions (expensive) cyclodextrins).

Page 9, lines 250. "four additional substrates". These additional substrates cover a limited space regarding functional group, size and structure (cyclic vs. linear) of substrates.

Page 9, lines 267. "SHC immobilization". Does it make sense to extract a membrane-associated

enzyme from the membrane to immobilize it on another support? Membrane association is some kind of immobilization. If side reactions do occur, it can be thought of working with membrane preparation after cell disruption.

Page 10, line 308: Protein production?

Page 11, line 337: was the impact of DMSO on SHC activity tested (at DMSO concentrations present in the reaction)?

Page 12, references. The references are not all in the same format.

Supporting information

In general: Tables and figures should be understandable on their own for clarity, they require therefore some additional work. Graphs: minor ticks are sometimes missing, as are sometimes axes lines.

Table S1: "SD" (here and elsewhere) abbreviation must be clarified. "m.c." spelled out in the column header for clarity.

Table S2: geranylacetone drawing and nomenclature as mentioned above: E/Z?, m.c. must be spelled out in column header.

Supporting text to Fig S1 and S2: substrate naming. Substrate and product were extracted. Were they included as neat compounds to the system, or extracted from a reaction? Should be clarified already here; legend to Figure S2 indicates biotransformation reactions. Is the conversion in the studied systems known? In other words, how can else be calculated % recovery of substrate and product? Requires clarification.

Figure S1 B: Y axis, % of what in what? If in the extract: should substrate and product in % sum up to 100? Requires clarification.

Figure S2 B: same question as above. A: Substrate and product: % of what with reference to what? Extraction yield, i.e. conversion known?

Figure S3, legend to Y axis? Substrate recovery?

Figure S5: Probably is meant protein loading of the immobilization carrier? Legend B, "recovered activity": better "molar conversion", same as title to Y-axis?

Figure S7: X axis: concentration (mM) of what (substrate probably, substrate is here geranylacetone) in what? Y axis: %? Needs better legend for clarity.

Table S4: substrate is geranylacetone, why not spell it out in the column header? Or clearly mention it in the title of the Table (bold part).

Table S5: same as above Table S4.

Figure S8: introducing colours here is not required. Using black/white is preferable as in other figures due to low complexity.

Figure S9: what means "relative conversion"? Is this simply conversion?

Table S8: as reading legends starts with reading the title it may be useful to have (here and elsewhere) the name of the substrate in the legend title when only one substrate is addressed. This allows for a better reading.

Figure S11: geranylacetone drawing? (as elsewhere in the document), and nomenclature?

GC chromatograms:

This information is always interesting. When provided in the supporting information straightforward reading must be ensured. In its present format it is barely readable (A4 format). Tables with retention times cannot be read and are useless. Chromatograms contain superfluous annotations (no relevant peaks are marked). The table should better not be integrated, and the chromatograms only be annotated with the relevant peaks including corresponding structure and retention time.

Geranylacetone: show instead of 4 chromatograms only one (full size on landscape format) with superimposed traces showing substrate, product, whole cells and spheroplasts biotransformations? Or 2 chromatograms: superimposed substrate and product, and 2 superimposed biotransformations?

Squalene/hopene and E,E-farnesol/drimenol, E,E-farnesylacetone/sclareoloxide, geranio/cyclogeraniol: add arrows to identify peak with structure? Drimenol instead of "Dimene"?

Page 18

GC-FID: quantification was made using dodecane as internal standard.

whole cells treated with SDS: on what basis was chosen the SDS:cells ratio of 0.05?

Page 19

Are detailed calculations required? It is assumed that calculations are made properly by the authors. Is it the readers task to cross check? Its formatting seems not appropriate, abbreviations used not necessarily clear, e.g. Experiment 1 and 2, c WT lyo? Concentration most probably, in mg/ml and M...

Reviewer #2 (Remarks to the Author):

This manuscript by Benítez-Mateos et al describes a novel strategy for obtaining stable and highly active biocatalysts that depend on the activity of membrane-associated proteins. In terms of applicability, the increase in squalene-hopene cyclase activity seems to be really promising compared to conventional methods. However, there are several issues regarding the preparation and manipulation of spheroplasts that need to be profoundly revised prior to publication, as the scaffold of the biocatalyst should be precisely characterized. In my opinion the authors may need assistance from experts in the field of membrane protein biochemistry as well as cell fractionation.

Major points

1. Spheroplast preparation. The authors state that the outer membrane has been removed, referring to Giannini et al, Prot Sci 2019, and Hobb et al, Microbiology 2018, for the protocol of spheroplast preparation. However, in the first work it has been clearly shown that spheroplasts retain the outer membrane. In fact, spheroplasts were used as permeabilized cells to measure the activity of an outer membrane protein. On the other hand, spheroplasts have been traditionally used as starting materials for the preparation of bacterial inner and outer membranes by isopycnic gradient (Osborn & Munson, Methods Enzymol. 1974). In this sense, the present work should include electron microscopy of spheroplasts, instead of optical microscopy, to visualize inner and outer membranes. In addition, the supernatant obtained after spheroplast preparation is the bacterial periplasm, and that should be clarified.

2. Protein expression. In the methods section the authors state that they use a pET22b(+) vector, which is ApR. Is it correct then to use of kanamycin?. Second, the SDS-PAGE shown in Figure S6 lacks a negative control without induction of protein expression. This should be included in order to identify AacSHC protein from the gel. In addition, the supernatant fraction (that would correspond to the periplasmic fraction) exhibits a protein pattern similar to whole cell. It is recommended to include a cytoplasmic contamination control, via Western-blot detection of a cytoplasmic protein

for instance, as a mean to discard cell lysis during spheroplast preparation.

3. Spheroplast manipulation. Spheroplasts lack the murein layer due to lysozyme treatment. In this way, unless the media are isotonic, spheroplasts will lyse, and this may have occurred in the current work. How can the authors be sure that they were actually working with intact spheroplasts and not with lysed cells or cell debris?. Furthermore, as before, electron microscopy of cross-linked spheroplasts should be performed to assess whether the picture shown in Figure 3A is correct.

4. In case this procedure is scaled up to industrial level, is it 1 mg/mL of lysozyme still affordable?

5. CHAPS is a detergent that solubilizes membrane proteins in detergent micelles, and not forming impermeable liposomes or membranes. The sentence "...that the diffusion through the membrane mimic CHAPS is still limited in the overall biotransformation" should be rewritten to avoid potential conceptual misunderstandings.

6. In the last section of Results, the authors point to the plastic polymer as responsible for the loss of substrate/product. Did the authors use a different flow system lacking that problem in the former experiments? Please clarify.

Minor points

-Table 1. It should say molar conversion instead of molar conversion

-Figure legends should define what error bars error bars represent (standard deviation, standard error of the mean, etc) and state the number of experimental replicates.

Reviewer #3 (Remarks to the Author):

The presented development and use of spheroplasts as carriers of squalene-hopene cyclase (SHC) is a successful example of the implementation of membrane-bound enzymes in industrially relevant biocatalytic production steps. Spheroplasts increased the bioavailability of the enzyme by removing the outer membrane and reducing the need for additives. Furthermore, spheroplasts crosslinking offers an innovative approach for the long-term use of membrane-bound enzymes, and lyophilization was found to be an efficient approach for the storage. However, the long-term use of crosslinked spheroplasts was not confirmed in a continuous biocatalytic process due to the substrate adsorption on the tubes.

Comments and suggestions for improvement:

1. The hypothesis that "the enhanced mass transfer that takes place in continuous flow reactors could benefit the SHC catalytic rate" is misleading and needs further clarification. Continuous flow reactors do not per se improve mass transfer. This could be stated only for the microreactors, where diffusion and mixing times are highly improved compared to conventional reactors. Apart from the mass transfer limitations by the cell wall, there is no evidence of the effect of mass transfer on the reaction in the homogeneous system. Please elaborate on this.
2. The structure of geranyl acetone in Scheme 1, and in tables in SI could be improved. The denomination of this compound is not consistent throughout the text.
3. The authors use protein content to calculate specific productivity (Table 1), which is defined per mass of the enzyme. Can authors comment on that?
4. Figure 2 and the corresponding text: abbreviations WC and WT are not clarified in the main article.
5. Please check the legend in Fig. 3C.
6. Please specify the whole names before the first abbreviation (E. coli, SDS etc).
7. In several figures, relative conversions are presented. However, it is not clearly stated relative to what these values are calculated and what is the maximal conversion obtained.
8. It is not clear how the crosslinked spheroplasts were retained in the flow reactor. What was the

size of the obtained CLSs and how they were packed in the column?

9. The reaction rate of biotransformations catalyzed by spheroplasts immobilized in various hydrogel beads is affected by the size of the beads or pieces, which are not specified.

10. It would be good to compare immobilization yields and efficiency for all tested immobilization approaches.

11. English language and typos could be improved (e.g. p.5 line 153: conversion, p.11, line 347: 4 mgg 1, p17 of SI:...with treated with...).

12. References: specify all authors (not et al.), use abbreviations for all journals, use italics for microorganisms, and unify the use of capital letters in titles.

REVIEWER COMMENTS

(Answers by authors in blue)

Spheroplasts preparation boosts the catalytic potential of a terpene cyclase

Ana I. Benítez-Mateos, Andreas Schneider, Eimear Hegarty, Bernhard Hauer*, and Francesca Paradisi*

Reviewer #1 (Remarks to the Author):

The authors address the use of spheroplasts in squalene hopene cyclase-catalysed reactions. These enzymes have attracted the attention of chemists and biochemists for several decades because unlike other terpene cyclases they do not require phosphorylated precursors, but proceed via a Brønsted acid catalysis-driven mechanism. Many reports are available showing the plasticity of these enzymes and their ability to be evolved through enzyme engineering resulting into tailor-made biocatalysts with modified substrate specificity and/or product selectivity. A recent example showed as well that these enzymes can be evolved to biocatalysts usable under process-relevant conditions in the field of flavours and fragrances.

- What are the noteworthy results? SHC enzymes are associated to the inner membrane of bacterial membranes e.g. when produced in *E. coli* for the use in biocatalytic reactions: the substrate has to access the enzyme through both outer and inner membranes. Cell permeabilization is a technique that is often addressed in the field of biocatalysis, whereas as the authors do mention carrying out biocatalytic reactions with spheroplasts is generally not considered. The authors demonstrate the advantage of using spheroplasts over whole cells (*E. coli* cells producing an SHC enzyme): the use of spheroplasts allows for a significant increase in activity. The authors introduce also the new concept of cross-linked spheroplasts and spheroplasts immobilization for the use in flow catalysis, flow reactions being of increasing interest in the field of biocatalysis.

- Will the work be of significance to the field and related fields? How does it compare to the established literature? The advantage of using of spheroplasts over whole cells reported in the present study may trigger an increase in interest for this biocatalyst formulation in biocatalytic reactions/processes in general. Novelty is addressed: reports on use of spheroplasts in biocatalytic reactions are rare/inexistent.

At this early stage of investigations is perhaps less evident the link between the use of spheroplasts and biocatalysis in flow. As mentioned by the authors more investigations must be done in this direction to confirm their applicability when immobilized and in flow reactions. Using SHC enzymes as target for such investigations is a difficult task due not only to the SHC enzyme itself, which is membrane-associated and handling water-insoluble terpenoid substrates, but also due to the currently used flow catalysis setup: reactivity/adsorption of terpenoids with plastic materials.

- Does the work support the conclusions and claims, or is additional evidence needed? The work supports the conclusions and claims regarding the better activity when considering SHC catalysis in the context of spheroplasts. It surely would have been interesting in this regard to extend/diversify the range of sub-strates used outside those tested and listed in Figure 2, which appear to be limited to squalene (AacSHC prototype substrate), C10 and C15 terpene-alcohols and -ketones. Using substrates as listed e.g. in Syren et al (2016), *Current Opin. Struct. Biol.* 41, 73-82 would have strengthened further the view of a broad applicability of the methodology, extending also the functional groups of substrates to aldehydes, acids, or epoxydes with epoxygeraniol, or aromatic substrates, cyclic compounds such as pinene, industrially-relevant substrates like homofarnesic acid, homofarnesol, and citronellal, substrates for the synthesis of sclarelolid, (-)-Ambrox and menthol,

- Are there any flaws in the data analysis, interpretation and conclusions? Do these prohibit publication or re-quire revision? Some points remain open, which after clarification/revision would turn this good into an excellent peace of work (de-tails below in “additional remarks”).

- Is the methodology sound? Does the work meet the expected standards in your field? Yes regarding the use of spheroplasts applied to SHC catalysis for boosting activity. Regarding biocatalysis in flow, it is not surprising that the use of (certain) plastic materials is an issue when working with terpenes/terpenoids in general. But the authors recognized the limit of the system and the urgency of adapting the usually used flow systems for working with terpenoids. This is an incentive to evolve further the design of flow biocatalysis setups.

- Is there enough detail provided in the methods for the work to be reproduced? Yes

Shall the comments I have below be addressed by the authors, I think this manuscript would be suited for publication in Nature Communications.

Comments to authors:

Manuscript

Title: It is possibly preferable to name the enzyme the authors work with directly in the title, i.e. “squalene hopene cyclase” and not using the general term “terpene cyclase”. Other terpene cyclases are not addressed in the submitted work. The question is if spheroplasts are relevant to working with terpene cyclases other than SHCs or oxidosqualene cyclases as these enzymes will use activated compounds as their substrates, which for my understanding are produced within the cells as part of their metabolism in biosynthesis processes, correct?

In agreement with the reviewer, the title has been amended as follows: Spheroplasts preparation boosts the catalytic potential of a squalene hopene cyclase.

The spheroplast biocatalyst can be an easy and smart solution for 1) membrane-bound enzymes and 2) highly hydrophobic substrates/products. In case that the substrate is produced within the microbial cell, the outer cell membrane would not play a role for the substrate access limitations to the enzyme. However, if the product is highly hydrophobic, the removal of the outer membrane could be an advantage for the product release to the reaction media. Moreover, a positive effect could be expected for other membrane-bound enzymes, as the protein structure could be maintained and therefore their stability.

Page 2 lines 38-42

The statement suggests an obligation/advantage of purifying membrane-associated/membrane-bound enzyme for efficient use in biocatalytic reactions, which is a difficult and costly task, questioning as mentioned enzyme stability when purification. But: is purification of such enzymes indeed required? Membrane-associated enzymes can be considered as “immobilized” per se. If using whole cells is an issue due to side reactions, then working with a membrane fraction may possibly solve this problem. There is to my understanding no need to justify with this statement the present work, which introduces the novel concept of using spheroplasts in biocatalytic reactions for boosting activity.

Depending on the substrate that is intended to cyclize, the “purification” of these enzymes may be needed. According to Siedenburg *et al.* 2011 “*Apparently, the outer membrane of E. coli is impermeable for squalene. Therefore, cell extracts or purified enzyme has to be used for determination of cyclase activity*”. This statement agrees with our results in Fig. 2., where we found the biggest difference in terms of TOF between the free enzyme (which is composed of cell extracts) and the whole cells.

To clarify our statement, we have amended the sentence in lines 39-40.

Page 2 lines 46-49

Cell membrane/diffusion barrier. If it is well understood that transporter enzymes would help for the transport of water-soluble compounds through the lipophilic cell membrane, would such transporters indeed allow for a better access of lipophilic substrates to the inner-membrane-associated SHC enzyme? These substrates would certainly well dis-solve/diffuse into the cell membrane. Would it be better for this reason to state here that lipid bilayers act rather as a sequestering agent for these compounds (as mentioned later e.g. Page 4, lines 103-107) rather than speaking of a diffusion barrier for lipophilic compounds?

Yes, transporters allow for a better access of lipophilic substrates. Julsing *et al.* 2012 reported the role of the outer membrane protein AlkL as a transport facilitator enabling a better access to the substrate. Nevertheless, the authors also reported "*The outer membrane of microbial cells forms an effective barrier for hydrophobic compounds, potentially causing an uptake limitation for hydrophobic substrates*". Moreover, as already mentioned, Siedenburg *et al.* 2011 also described a diffusion barrier imposed by the outer membrane of *E. coli*: "*Apparently, the outer membrane of E. coli is impermeable for squalene*". To sum up, there is a dual effect of the outer membrane acting as a sequestering agent and as a diffusion barrier as well. Therefore, we have now mentioned both effects in page 2, lines 46-49.

Page 3, Scheme 1. The representation of the mixture of geranylacetone isomers may not be the best (same later on in the manuscript and supplementary information). Should the two isomers better be shown for clarity? Nomenclature: why introducing t/c (cis/trans) as the manuscript mentions often the E-isomer (also in legend to scheme 1). Why not better stick for consistency/clarity to this E/Z nomenclature throughout the manuscript and not use t/c?

We agree with the reviewer's suggestion. Therefore, we have changed the nomenclature t/c and stick to the E/Z throughout the manuscript and supplementary information. Moreover, scheme 1 shows now both isomers for better clarity.

Page 3, lines 93-95. Is it possible that entrapment adds an additional barrier to the diffusion of substrates to the enzyme?

Initially, we had the same concern as the reviewer. However, Figure S11A shows similar amounts of substrate/product after the biotransformations with spheroplasts and the biotransformations with entrapped spheroplasts. Therefore, the alginate entrapment is not contributing substantially to hamper the substrate diffusion.

Page 4, lines 112-115. Flow biotransformations are carried out in a mixture of buffer and solvents. Did the authors test the influence of solvent addition on SHC activity, e.g. if the solvents used inhibit activity?

Yes, after the experiment depicted in Fig. S3, the activity of the immobilized SHC was tested in batch to confirm that the biphasic system had no influence on the enzyme activity. This information has been added to the figure caption.

Page 4, 2nd paragraph. Do the % conversion values reported really allow to include wording like "activity improvement" or "effect on conversion" as all the values are between 4 and 7%, probably being within the error margin inherent to the system?

The reviewer is correct, the differences between 4 and 7% are minimal. Therefore, we changed the sentence: "*Likewise, increasing the retention time and temperature **had a minimal effect** (8%*

conversion)” by “Likewise, increasing the retention time and temperature **did not show any significant improvement** (8% conversion)”.

Page 4, line 124. Are proteins expressed in bacteria? This wording became common language, but isn't it of general understanding that genes are expressed resulting into protein/enzyme production.

Agreed. The sentence has been modified with a more appropriate terminology: “[...]cytoplasmic membrane when **produced** in bacteria”

Page 4, line 129/131. From the text it is not straight forward how biotransformation yield and productivity are related. Could biotransformation yield here better be named conversion as it is in Table 1?

To avoid confusion, the word “yield” has been removed and only “conversion” is used to refer to the % of product formed. Productivity has been only used to indicate g/L of product.

Page 4, line 133. Did whole cells retain substrate and product indeed, or could have an alternative extraction method allow for a better extraction?

In our experience, whole cells retained both substrate and product. To test this, we compared the extracted substrate and product from the reaction mixture containing cells with another identical reaction mixture without cells. This control has now been added to the Fig. S2B for clarity.

Regarding the extraction method, we also tested toluene in this work obtaining similar extraction results (Fig. S2). Previously, other solvents (i.e. MTBE) have been tested in the lab of Prof. Hauer but no advantages have been observed in the extraction step. Therefore, we have used ethyl acetate:cyclohexane (1:1) which offer good results due to the different polarities of the solvents.

Page 5, first paragraph. Spheroplasts storage: how well are spheroplasts storable at 4°C?

Spheroplasts can be stored for at least 2 weeks at 4°C maintaining enzymatic activity. This information has been added to the manuscript.

Page 5, Table 1. Why not indicating for clarity in the Table column header “molar conversion (%)” in full instead of using a not trivial abbreviation (m.c.)? This applies throughout Tables/Figures in the manuscript and supplementary information (wherever the abbreviation is not spelled out). Productivity: are the reported values not micromol product/mg enzyme: change unit in table header for straightforward reading?

We have changed the terminology for a better clarity as requested by the reviewer.

Specific activity: calculated on 24h. Isn't it better to calculate this value over the linear reaction rate as possibly a plateau might be reached at different timepoints depending on the biocatalyst formulation considered? How do “productivity (24 h)” and “specific activity calculated in 24 h biotransformations” correlate? This must be clarified.

The reviewer is right. Specific activity should not be calculated on 24 h when the reaction rate has reached a plateau. Therefore, we have decided to remove that column from the table 1. Moreover, the results shown in the column “biocatalyst productivity in 24h” provided already similar information.

Page 6. It would have been relevant to extend the range of examples of substrates (size, functional groups) to generalize the concept of “spheroplasts as biocatalysts for the efficient cyclization of terpenes”.

We agree with the reviewer and added four more examples including aldehyde and carboxylic acid functionalities as well as industrially relevant cyclization of homofarnesol towards high-value fragrance ingredient (-)-Ambroxide.

Page 7, lines 182-184. Spheroplasts use at higher cell concentration. It looks like increasing substrate concentration has a negative effect on catalytic activity (Figure S8), the relative conversion decreases with increasing substrate concentration. Is this correct, and is there an explanation for this? Could this be a general rule? Was this expected? Here the use of whole cells seems to be of an advantage. Do the reported values in the text really match the ones displayed (conversion) in Figure S8: better conversion with agarose entrapment for cyclization of 3 to 4, slightly better conversion with alginate entrapment for cyclization of 7 to 8?

This observation is correct and was expected by the authors. Our hypothesis behind this effect is that the cell membrane acts a “organic phase” for hydrophobic terpenes, which entails changes in the membrane fluidity (see Belin, B. J.; Busset, N.; Giraud, E.; Molinaro, A.; Silipo, A.; Newman, Di. K. Hopanoid Lipids: From Membranes to Plant-Bacteria Interactions. *Nat. Rev. Microbiol.* 2018, 1 (5), 304–315. and Mendanha, S. A.; Alonso, A. Effects of Terpenes on Fluidity and Lipid Extraction in Phospholipid Membranes. *Biophys. Chem.* 2015, 198, 45–54. for more detailed information). Furthermore, we believe that the squalene-hopene cyclase is regulated by the membrane fluidity (see *ChemRxiv*, 2022. doi: 10.26434/chemrxiv-2022-xwcqj for some kinetic data). The whole cell preparation provides more “organic phase” by the additional membrane to widespread a membrane distortion effect. As a result, the whole cells perform better at higher substrate concentrations. The Hauer Lab is currently working on this intricate mechanism and will provide new insights soon.

We agree with the reviewer and change the wording regarding the agarose and alginate entrapment.

Why was then alginate entrapment used for entrapment for cyclization of 1 as the result suggest that best method for entrapment could depend on substrate considered? Is this conclusion correct?

Yes, the reviewer is right. That is the reason why we tested different entrapment methods for cyclization of 1 (Fig. S11), and we found alginate to be the best entrapment method for that substrate.

Page 7, spheroplasts entrapment in hydrogel beads. Does entrapment add a new barrier to substrate transfer to the enzyme?

As Figure S11 shows, alginate entrapment does not add a new barrier to the substrate because the same conversion was obtained compared with the non-entrapped spheroplasts. In both cases, the maximum conversion for the cyclization of the *E*-isomer (50%) was almost obtained.

Page 8, SHC reactions in flow system. Reactivity of plastic materials with terpenoids is known and requires some precautions.

We agree, and other alternatives (possibly metal tubings etc) will be explored in the future, but the set up is not easily changed with our current system.

Page 9, line 246. Is it the case that ref 33 states that SHC reactions have remained a challenging task

until now? Isn't it rather the case that these enzymes were simply to date mainly studied for their plasticity/evolvability rather than for their applicability at large scale (process-suitability)?

The ref 33 does not state explicitly that SHC reactions have remained a challenging task until now. The citation to the ref 33 here refers to "*This is particularly relevant for SHC reactions which are getting great attention in the flavour industry to synthesize enantiopure cyclic terpenoids*". To avoid misunderstandings, the sentence has been shortened.

It is true that SHC enzymes have been mainly studied for the plasticity/evolvability, but also their applicability in production scale has been studied as it is described in ref 33 and ref 10.

Page 9, lines 246-249. True: no additives added to the biotransformation itself. But the preparation of spheroplasts requires additional preparation compared to whole cells, involving sucrose, NaCl, EDTA and lysozyme. Is it therefore possible to directly conclude in general on the more cost-efficient and sustainable alternative of using spheroplasts vs. whole cells? This can, but must not, be the case depending on the required "additives" (Table S8 which is referred to only mentions (expensive) cyclodextrins).

The preparation of spheroplasts is a simple and quick process that takes less than a couple of hours. Moreover, the reagents needed to prepare the spheroplasts are all together cheaper than the addition of cyclodextrins: 15,300 CHF/Kg. While it is true that the costs depend on the type of additives (SDS: 367 CHF/Kg), the continuous addition of any surfactant or solubilizing agent to the reaction mixture on large scale will definitively increase the process costs. Similarly, the addition of surfactants or solubilizing agents will raise the waste production and complicate the product isolation. Therefore, it is possible to conclude that the use of spheroplasts in general is a more cost-efficient and sustainable alternative, especially when they can be reused.

Page 9, lines 250. "four additional substrates". These additional substrates cover a limited space regarding functional group, size, and structure (cyclic vs. linear) of substrates.

We agree with the reviewer and added four more examples including aldehyde and carboxylic acid functionalities as well as industrially relevant cyclization of homofarnesol towards high-value fragrance ingredient (-)-Ambroxide.

Page 9, lines 267. "SHC immobilization". Does it make sense to extract a membrane-associated enzyme from the membrane to immobilize it on another support? Membrane association is some kind of immobilization. If side reactions do occur, it can be thought of working with membrane preparation after cell disruption.

We have observed that not all substrates behave the same and in fact substrate 9 only works with the extracted enzyme. It is true that membrane association is in fact a form of immobilisation, but we were hoping to find a material which would not have the same impact as the membrane on the adsorption of substrate/product in many cases.

The new Figure S9 depicting the molar conversion of the results in Fig. 2 has been added for a better understanding.

Page 10, line 308: Protein production?

Yes, we have changed the term to make it more adequate.

Page 11, line 337: was the impact of DMSO on SHC activity tested (at DMSO concentrations present in the reaction)?

The DMSO concentration used in the reactions was only 1% and did not affect the SHC activity nor stability.

Page 12, references. The references are not all in the same format.

The references have been amended accordingly.

Supporting information

In general: Tables and figures should be understandable on their own for clarity, they require therefore some additional work. Graphs: minor ticks are sometimes missing, as are sometimes axes lines.

Additional information has been added to the figures/tables captions as well as the missing minor ticks.

Table S1: "SD" (here and elsewhere) abbreviation must be clarified. "m.c." spelled out in the column header for clarity.

"SD" abbreviation has been clarified "m.c." has been changed by "molar conversion" everywhere in the manuscript and the supporting information.

Table S2: geranylacetone drawing and nomenclature as mentioned above: E/Z?, m.c. must be spelled out in column header.

The nomenclature suggested above (E/Z) has been adopted throughout the manuscript and the supporting information. Moreover, m.c. has been spelled out in column header.

Supporting text to Fig S1 and S2: substrate naming. Substrate and product were extracted. Were they included as neat compounds to the system, or extracted from a reaction? Should be clarified already here; legend to Figure S2 indicates biotransformation reactions. Is the conversion in the studied systems known? In other words, how can else be calculated % recovery of substrate and product? Requires clarification.

The correct nomenclature for the substrate **1E/Z** has been added to the Supporting text to Fig S1 and S2.

Good point about the substrate and production extractions which were not very clear. Now, we have specified that the substrate and product were extracted from the reactions in Fig.S1B-C and Fig. S2A. On the contrary, the results depicted Fig. S2B corresponds to an extraction test of the support alone (without cells) to verify the impact of the material in sequestering the substrate. The % recovery of substrate (Fig. S2B) was calculated in comparison with a solution containing only the substrate which has been also added to the graph for clarity. Moreover, the legend of the Y axis has been modified to "recovery (%)" for a better understanding.

Figure S1 B: Y axis, % of what in what? If in the extract: should substrate and product in % sum up to 100? Requires clarification.

Light grey bars correspond to the % of recovery of substrate and dark grey bars correspond to the % of recovery of product as it specified in the figure legend.

The recovery (%) of substrate and product were calculated by using standard curves of the substrate and product. The reviewer is right, substrate and product in % should sum up to 100. However, that is exactly the problem here. The recovery (%) of substrate and product do not sum up to 100% due to the high affinity of the substrate to the supports (Gx-Agarose and Gx-methacrylate) as well as the retention of the product inside the whole cells (Fig. S2B). Clarifications have been added to the figure caption. Moreover, the legend of the Y axis has been modified to “recovery (%)” for a better understanding.

Figure S2 B: same question as above. A: Substrate and product: % of what with reference to what? Extraction yield, i.e. conversion known?

Clarifications have been added to the figure caption. Fig. S2A shows the recovery (%) of substrate and product after the biotransformations by using standard curves of the substrate and product (similar to Fig. S1B). In the case of Fig. 2B, only the recovery (%) of substrate was calculated in comparison with a solution containing only the substrate which has been also added to the graph for clarity. Moreover, the legend of the Y axis has been modified to “Substrate recovery (%)” for a better understanding.

Figure S3, legend to Y axis? Substrate recovery?

Yes, the legend of the Y axis has been modified to “Substrate recovery (%)” for a better understanding.

Figure S5: Probably is meant protein loading of the immobilization carrier? Legend B, “recovered activity”: better “molar conversion”, same as title to Y-axis?

Yes, we have amended the caption as requested.

Figure S7: X axis: concentration (mM) of what (substrate probably, substrate is here geranylacetone) in what? Y axis: %? Needs better legend for clarity.

X and Y axes have been modified as suggested.

Table S4: substrate is geranylacetone, why not spell it out in the column header? Or clearly mention it in the title of the Table (bold part).

Done.

Table S5: same as above Table S4.

Done.

Figure S8: introducing colours here is not required. Using black/white is preferable as in other figures due to low complexity.

Colours have been changed.

Figure S9: what means “relative conversion”? Is this simply conversion?

Yes, we did in fact mean conversion. For consistency, we had maintained the same terminology as in previous publications (i.e. Schneider *et al.* 2021 *Angew. Chem. Int. Ed.*) where the relative conversion

was calculated as: $\text{product} / (\text{substrate} + \text{product}) \times 100$. However, since whole cells can sequester the substrate/product, in this work we have calculated conversion as % of product by using standard curves of the product. Therefore, we have used only “molar conversion” or “conversion” in the revised versions of the manuscript and supporting information.

Table S8: as reading legends starts with reading the title it may be useful to have (here and elsewhere) the name of the substrate in the legend title when only one substrate is addressed. This allows for a better reading.

Done.

Figure S11: geranylacetone drawing? (as elsewhere in the document), and nomenclature?

In agreement with the reviewer's suggestion, we have drawn the reaction scheme with the two isomers in the scheme 1 of the main manuscript. For the sake of clarity and to avoid unnecessary repetitions, we have removed the schemes of the cyclization of geranyl acetone 1*E/Z* from the supporting information, but we have used the suggested nomenclature for the substrate (1*E/Z*) and the product (2*E*).

GC chromatograms:

This information is always interesting. When provided in the supporting information straightforward reading must be ensured. In its present format it is barely readable (A4 format). Tables with retention times cannot be read and are useless. Chromatograms contain superfluous annotations (no relevant peaks are marked). The table should better not be integrated, and the chromatograms only be annotated with the relevant peaks including corresponding structure and retention time.

Geranylacetone: show instead of 4 chromatograms only one (full size on landscape format) with superimposed traces showing substrate, product, whole cells and spheroplasts biotransformations? Or 2 chromatograms: superimposed sub-strate and product, and 2 superimposed biotransformations? Two superimposed chromatograms have now been included as suggested by the reviewer: only relevant peaks are shown, the table has been removed, and molecule structure and retention times are also shown.

Squalene/hopene and E,E-farnesol/drimenol, E,E-farnesylacetone/sclareoloxide, geranio/cyclogeraniol: add arrows to identify peak with structure ? Drimenol instead of “Dimene”?

Lines were added to the chromatograms. Typing errors were changed.

Page 18

GC-FID: quantification was made using dodecane as internal standard. whole cells treated with SDS: on what basis was chosen the SDS:cells ratio of 0.05?

The ratio was chosen as it was evaluated by Eichhorn *et al.* in an extensive study regarding the whole cell biotransformation of homofarnesol. Please see Eichhorn, E.; Locher, E.; Guillemer, S.; Wahler, D.; Fourage, L.; Schilling, B. Biocatalytic Process for (-)-Ambrox Production Using Squalene Hopene Cyclase. *Adv. Synth. Catal.* 2018, 350 (12), 2339–2351.

Page 19

Are detailed calculations required? It is assumed that calculations are made properly by the authors. Is it the readers task to cross check? Its formatting seems not appropriate, abbreviations used not necessarily clear, e.g. Experiment 1 and 2, c WT lyo? Concentration most probably, in mg/ml and M...

The detailed calculations have been removed from the supporting information as requested (occasionally reviewers ask to include details of even simple calculations...)

Reviewer #2 (Remarks to the Author):

This manuscript by Benítez-Mateos et al describes a novel strategy for obtaining stable and highly active biocatalysts that depend on the activity of membrane-associated proteins. In terms of applicability, the increase in squalene-hopene cyclase activity seems to be really promising compared to conventional methods. However, there are several issues regarding the preparation and manipulation of spheroplasts that need to be profoundly revised prior to publication, as the scaffold of the biocatalyst should be precisely characterized. In my opinion the authors may need assistance from experts in the field of membrane protein biochemistry as well as cell fractionation.

Major points

1. Spheroplast preparation. The authors state that the outer membrane has been removed, referring to Giannini et al, Prot Sci 2019, and Hobb et al, Microbiology 2018, for the protocol of spheroplast preparation. However, in the first work it has been clearly shown that spheroplasts retain the outer membrane. In fact, spheroplasts were used as permeabilized cells to measure the activity of an outer membrane protein. On the other hand, spheroplasts have been traditionally used as starting materials for the preparation of bacterial inner and outer membranes by isopycnic gradient (Osborn & Munson, Methods Enzymol. 1974). In this sense, the present work should include electron microscopy of spheroplasts, instead of optical microscopy, to visualize inner and outer membranes. In addition, the supernatant obtained after spheroplast preparation is the bacterial periplasm, and that should be clarified.

The reviewer is right, Giannini *et al.* affirmed the spheroplasts produced with their protocol maintained the outer membrane. It could be that the lysozyme-treatment applied by Giannini *et al.* was not sufficient to remove the outer cell membrane (e.g. much lower EDTA concentration, 0.1 mM EDTA). However, "spheroplasts" are widely defined as *gram-negative bacterial cells lacking the outer membrane*. Hence, there may be a disagreement on the definition of "spheroplasts". Please, see some of the statements cited:

- Figueroa *et al.* 2018 JOVE: "*Spheroplasts and protoplasts differ from normal bacteria, most notably in their lack of an outer cell wall [...]*."
- Hobb *et al.* 2009 Microbiology: "*It has been shown that during spheroplast formation, outer membranes of E. coli and Pseudomonas are broken by the lysozyme and EDTA and form complex-coiled structures, while the cytoplasmic membrane remains intact (i.e. the spheroplast) [...]*." and "*The method described here was based on the work of Osborn & Munson (1974) and Hill & Silence (1997) and produces rapid and efficient spheroplasting of bacteria while the outer membrane tends to peel away during spheroplast formation (Osborn & Munson, 1974).*"
- Sun *et al.* 2014 Biophysical Journal. "*Experimentalists have made use of spheroplasts, the cells from which the outer membranes have been removed, for patch-clamp, fusion, and other experiments and also for antibiotic studies*".

Therefore, we have removed the Giannini citation from the manuscript. As requested by the reviewer, we have performed TEM analysis of the whole cells and spheroplasts which confirm the removal of the outer membrane of spheroplasts following the protocol reported in this work. We are really grateful for this suggestion as it allowed us to properly visualize what we were generating. The TEM images have replaced the previous optical microscopy images (that have been moved to the ESI) in Fig.1. We would like to stress the fact that the outer membrane removal is still partial, as it has been previously reported, meaning that the spheroplast fraction may still contain some remaining whole cells and some cell debris. This has been also clarified in the revised version.

In addition, we have specified the content of the supernatant fraction in Table 1.

2. Protein expression. In the methods section the authors state that they use a pET22b(+) vector, which is ApR. Is it correct then to use of kanamycin?.

We apologize for the mistake, the reviewer is correct, Ampicillin was indeed used, and this has been now modified in the methods.

Second, the SDS-PAGE shown in Figure S6 lacks a negative control without induction of protein expression. This should be included in order to identify AacSHC protein from the gel.

A new SDS-PAGE showing the negative control without induction of protein expression. Has been added to Figure S6.

In addition, the supernatant fraction (that would correspond to the periplasmic fraction) exhibits a protein pattern similar to whole cell. It is recommended to include a cytoplasmic contamination control, via Western-blot detection of a cytoplasmic protein for instance, as a mean to discard cell lysis during spheroplast preparation.

The reviewer makes a very good point. We have now performed a different control which we believe gives a very good indication of the efficiency of process as we do not have at hand Western-blotting reagents. As cytoplasmic control, we have produced sGFP (superfolded green fluorescent protein) which is a soluble protein in the cytoplasm and can be easily monitored by fluorescence. Then, the *E. coli* cells harbouring sGFP were used to prepare spheroplasts and the fluorescence in the supernatant was measured during the process (see supporting methods). This experiment shows that about 35-45% of the initial whole cells can be recovered spheroplasts following the EDTA-lysozyme protocol reported in this work, while the rest of whole cells might be lysed (Fig. S7). This suggested experiment enabled us to better correlate these results with the previous data reported in Table 1, where in fact only 35% (0.28 mg) of AacSHC was also found in the spheroplasts compared to the initial protein content (0.8 mg) in the whole cells.

This is certainly indicative that some further optimization is needed to increase the amount of spheroplasts obtained, ideally making the process less harsh (perhaps reduced incubation times?) for future applications. We think that in this case, we can already clearly show the advantages of spheroplasts biocatalysts, and the new results and discussion have been added to the main manuscript.

3. Spheroplast manipulation. Spheroplasts lack the murein layer due to lysozyme treatment. In this way, unless the media are isotonic, spheroplasts will lyse, and this may have occurred in the current work. How can the authors be sure that they were actually working with intact spheroplasts and not with lysed cells or cell debris?. Furthermore, as before, electron microscopy of cross-linked spheroplasts should be performed to assess whether the picture shown in Figure 3A is correct.

As previously described, sucrose (10%) is added during the preparation of spheroplasts to improve their stability and avoid lysis during the EDTA-lysozyme treatment. After the preparation of spheroplasts, the ionic strength of the buffer is kept identical to avoid lysis. But the reviewer makes a very good point and we have performed TEM analysis of the CLS confirming the spheroplast integrity after crosslinking with glutaraldehyde (Fig. S12C), while this is clearly not the case when we cross-link with PEI, even though in both cases we observe a macroscopic change in appearance with larger aggregates, as well as reusability of both preparations.

4. In case this procedure is scaled up to industrial level, is it 1 mg/mL of lysozyme still affordable?

We shared the same concern as the reviewer, in fact we even considered producing our own lysozyme. However, the application of lysozyme to the spheroplast preparation is quite affordable. According to the prices on Sigma Aldrich (20/06/2022), starting with 100 mg of cells, we need 1 mg of lysozyme which costs 0.0357 CHF to prepare the spheroplasts. The resulting

spheroplasts can be used as a biocatalyst for a 10 mL reaction. If we scale this up to, for example, 100 L of reaction, one would need 1 Kg of cells whose lysozyme cost would be only 357 CHF. This would be well absorbed by the high value of some of the products we can now easily generate.

5. CHAPS is a detergent that solubilizes membrane proteins in detergent micelles, and not forming impermeable liposomes or membranes. The sentence “..that the diffusion through the membrane mimic CHAPS is still limited in the overall biotransformation” should be rewritten to avoid potential conceptual misunderstandings.

We agree with the reviewer. Therefore, the sentence has been reworded.

6. In the last section of Results, the authors point to the plastic polymer as responsible for the loss of substrate/product. Did the authors use a different flow system lacking that problem in the former experiments? Please clarify.

Unfortunately, we could not explore other alternative tubing materials due to their limited commercial availability for flow-reactor systems. Further investigations into alternative tubing material will be done in the future. This information has been added to the manuscript for clarification.

Minor points

-Table 1. It should say molar conversion instead of molar conversion

The typo has been corrected.

-Figure legends should define what error bars error bars represent (standard deviation, standard error of the mean, etc) and state the number of experimental replicates.

The information has been added.

Reviewer #3 (Remarks to the Author)

The presented development and use of spheroplasts as carriers of squalene-hopene cyclase (SHC) is a successful example of the implementation of membrane-bound enzymes in industrially relevant biocatalytic production steps. Spheroplasts increased the bioavailability of the enzyme by removing the outer membrane and reducing the need for additives. Furthermore, spheroplasts crosslinking offers an innovative approach for the long-term use of membrane-bound enzymes, and lyophilization was found to be an efficient approach for the storage. However, the long-term use of crosslinked spheroplasts was not confirmed in a continuous biocatalytic process due to the substrate adsorption on the tubes.

Comments and suggestions for improvement:

1. The hypothesis that “the enhanced mass transfer that takes place in continuous flow reactors could benefit the SHC catalytic rate” is misleading and needs further clarification. Continuous flow reactors do not per se improve mass transfer. This could be stated only for the microreactors, where diffusion and mixing times are highly improved compared to conventional reactors. Apart from the mass transfer limitations by the cell wall, there is no evidence of the effect of mass transfer on the reaction in the homogeneous system. Please elaborate on this.

As stated by Tamborini et al. 2018: “Flow processing has the potential to accelerate biotransformations due to enhanced mass transfer, [...]”. Nevertheless, we agree with the reviewer that the effect on the mass transfer that happens in mesoreactors is clearly poorer than the extremely effective heat and mass transfer that takes place in microreactors, and it cannot be used as a blank statement.

Then, the sentence has been rephrased accordingly.

2. The structure of geranyl acetone in Scheme 1, and in tables in SI could be improved. The denomination of this compound is not consistent throughout the text.

In agreement with the reviewer, the structure of geranyl acetone has been improved in Scheme 1. For a better clarity, we have now used the isomer denomination (E/Z) throughout the text and the supporting information.

3. The authors use protein content to calculate specific productivity (Table 1), which is defined per mass of the enzyme. Can authors comment on that?

The Table S1 showed “productivity in 24h” and “specific activity”. However, we think that as mentioned also by another reviewer, specific activity should not be calculated over 24 h when the reaction rate might have reached a plateau. Therefore, we have decided to remove that column from the table 1. Moreover, the results shown in the column “biocatalyst productivity in 24h” provided already similar information.

4. Figure 2 and the corresponding text: abbreviations WC and WT are not clarified in the main article.

Clarifications of the abbreviations have been added to the figure caption.

5. Please check the legend in Fig. 3C.

We are not sure what the reviewer refers to. It might be that the results for “spheroplasts control” and “CLS BDE” did not appear in the graph as the values were zero because it was not possible to reuse those biocatalysts. Due to the lack of crosslinking, those biocatalysts are lost in the flow-through during the filtration. This has been clarified in the caption.

6. Please specify the whole names before the first abbreviation (E. coli, SDS etc).

Done.

7. In several figures, relative conversions are presented. However, it is not clearly stated relative to what these values are calculated and what is the maximal conversion obtained.

This was noted also by another reviewer, we simply meant conversion but we had maintained the same terminology as in previous publications (i.e. Schneider *et al.* 2021 *Angew. Chem. Int. Ed.*) where the relative conversion was calculated as: $\text{product} / (\text{substrate} + \text{product}) \times 100$. However, since whole cells can sequester the substrate/product, in this work we have calculated conversion as % of product by using standard curves of the product. Therefore, we have used only “molar conversion” or “conversion” in the revised versions of the manuscript and supporting information.

8. It is not clear how the crosslinked spheroplasts were retained in the flow reactor. What was the size of the obtained CLSs and how they were packed in the column?

CLS are solid and heterogenous aggregates with a larger size (hundreds μm to a few mm) than the filter of the flow reactor (10 μm). In the new Fig. S12C, we have added a photograph and two microscopy images with more details on the size of CLS.

In addition, we have added more details to the methods section about how CLS (and all the other immobilized biocatalysts) were packed in the column.

9. The reaction rate of biotransformations catalyzed by spheroplasts immobilized in various hydrogel beads is affected by the size of the beads or pieces, which are not specified.

All the hydrogel beads were of similar size: 1-4 mm. This information has been added to the caption of the Fig. S11.

10. It would be good to compare immobilization yields and efficiency for all tested immobilization approaches.

A new table containing all the immobilization yields has been added to the supporting information.

11. English language and typos could be improved (e.g. p.5 line 153: conversion, p.11, line 347: 4 mgg 1, p17 of SI:...with treated with...).

We are sorry for the typos and we have corrected all those that we spotted in the revised versions of the manuscript and supporting information. English language has been edited by a native speaker.

12. References: specify all authors (not et al.), use abbreviations for all journals, use italics for microorganisms, and unify the use of capital letters in titles.

The references have been amended according to the *Nature* style (et al. has to be used, italics have been implemented for organisms, capital letters in titles have been unified, journal titles have been abbreviated correctly etc.).

REVIEWER COMMENTS

Reviewer #1 (Remarks to the Author):

The authors overall addressed the concerns raised in the first review round. Some comments to the new version below, requiring some additional modifications.

Figure 3: No abbreviations must be used in titles to axes: what is here m.c.? this is even not defined in the legend to figure 3! Authors should pay much attention to the consistency of Figures and legends to Figures: "A B C" in the graphs, but "a, b, c" in the legend. As mentioned earlier: a figure with its legend must be understood as standalone material, not being obliged to look to the text in the publication to find out what is what. Colour in Figure 3 C is not justified and should better be left out.

Page 10, line 283: "high value cyclization of homofarnesol": no meaning in particular as the product is not mentioned. The reader does not know what this is exactly about. Why not mentioning here the isomer of interest, as is for homofarnesoic acid later on in line 287. Assuming that not both homofarnesol isomers will lead to the compound of interest?

Page 12, line 359, preparation of spheroplasts. The preparation of spheroplasts requires sucrose, NaCl, EDTA, lysozyme.

Page 60, line 60, legend to Figure 1. "using detergents enhances mass transfer but increases costs and creates additional waste.

Looking at these two points, the question is still open what is best in terms of cost and waste production. In that view the author did not address the point raised earlier. The production of each of these ingredients has a cost and produces waste. The question remains open if it is less costly and waste generating to (i) produce the cells and engage them in a bioconversion where is only added some detergent, or (ii) produce the cells and run them through the preparation of spheroplasts requiring four ingredients prior to running the bioconversion. At what cost (price and waste is produced e.g. lysozyme, EDTA)? This is not necessarily clear. The point here is not to claim the authors are right or wrong in their conclusion regarding cost and waste. It is simply not possible to conclude on such a topic without having carried out a complete analysis regarding waste and cost of the 2 processes. This is why the authors should refrain of making such not well-founded but "cheap" statements. Having not carried out this analysis for making conclusions based on facts, it should be left at one's own discretion what is better rather than making subjective conclusions.

Page 6 line 187. Spheroplast preparation surpassed the reported optimal setup of whole cells treated with SDS. This is the author's result in the comparison they did. However, from where comes the judgement that the reported setup is "optimal", as claimed by the authors? It is here probably sufficient to refer to a "reported setup". Here also the authors make a subjective conclusion about what was published elsewhere.

Figure S9. I assume that with the substrates listed the authors have looked for some of the substrates at the conversion of only one particular isomer? As is indicated in Figure 2. Same care should then be taken here for the annotations of Fig S9 (13, 17?)

Reviewer #2 (Remarks to the Author):

The authors performed new experiments that were clear enough to answer all my previous concerns, and the manuscript was accordingly amended. At least in what regards spheroplast generation and manipulation, I think the work presented is correct.

Reviewer #3 (Remarks to the Author):

The authors have improved the manuscript considerably and have taken into account most of the reviewers' suggestions.

However, a few points have not yet been satisfactorily elaborated:

1. The response and corresponding changes regarding "the enhanced mass transfer that takes place in continuous flow reactors," which refers to the corresponding author's review paper, are not correct, especially since the reactor type is not specified. For example, a stirred tank reactor can be operated in either batch or continuous mode, and mass transfer is enhanced only by the stirring rate. Similarly, mass transfer in tubular reactors, spinning disk reactors, etc., depends on flow conditions and, for heterogeneous catalysts, also on packing (which determines channeling and pressure drop) and particle size and porosity (which determine internal mass transfer). The authors should consider continuous processing with immobilized biocatalysts as a step toward increasing the productivity of biocatalytic processes (process intensification), which is also related to the miniaturization of flow reactors (more on this topic in doi:10.3390/catal9030262 and doi:10.1016/j.cogsc.2021.100546).
2. The use of the term "enzyme" instead of "protein" or "preparation" in the reported concentrations or calculated productivities is not correct. Since the preparations are not purified enzymes, this should be carefully checked and corrected. Biocatalyst productivity in Table 1 is defined per enzyme but should be reported per protein, and biocatalyst concentrations in Figure 2 are defined as mg of enzyme per mL but should be reported in mg of protein, better yet, in U/mL. In the supporting text to Figure 4, the "enzyme" concentration is given, which should better be changed to "enzyme preparation", which is correctly used p.23 of SI to denote the partly purified preparation.
3. The problem with the legend in Figure 3C referred to only 2 results and 4 subjects in the legend. It was good to omit the two cases with zero values and provide this information in the caption. However, titles of Y-axes in Figures 3B and 3C should be consistent with the other graphs/tables and use "Molar conversion" and not the abbreviation m.c.
4. The authors have included more data and images of CLSs, which is welcome. However, Figure SI12B is not very informative in its current state (the CLSs are almost not visible), and the macroscopic image of Figure SI12C should also be enlarged to focus on the particles and less on the ruler to give a clearer picture of the CLSs.
5. It was good to compare immobilization yields of various immobilization techniques tested. Can you also compare the retained activities of obtained preparations?

REVIEWER COMMENTS

(Answers by authors in blue)

Spheroplasts preparation boosts the catalytic potential of a terpene cyclase

Ana I. Benítez-Mateos, Andreas Schneider, Eimear Hegarty, Bernhard Hauer*, and Francesca Paradisi*

Reviewer #1 (Remarks to the Author):

The authors overall addressed the concerns raised in the first review round. Some comments to the new version below, requiring some additional modifications.

Figure 3: No abbreviations must be used in titles to axes: what is here m.c.? this is even not defined in the legend to figure 3! Authors should pay much attention to the consistency of Figures and legends to Figures: "A B C" in the graphs, but "a, b, c" in the legend. As mentioned earlier: a figure with its legend must be understood as standalone material, not being obliged to look to the text in the publication to find out what is what. Colour in Figure 3 C is not justified and should better be left out.

The abbreviations have been replaced by "Molar conversion" following the nomenclature in the rest of the paper and ESI. Likewise, "A B C" in the graphs of Figure 1 and 3 have been replaced by "a, b, c" following the guidelines of Nature Communications. In addition, the colours in Figure 3C have been replaced by a grey scale.

Page 10, line 283: "high value cyclization of homofarnesol": no meaning in particular as the product is not mentioned. The reader does not know what this is exactly about. Why not mentioning here the isomer of interest, as is for homofarnesoic acid later on in line 287. Assuming that not both homofarnesol isomers will lead to the compound of interest?

We have added the name of the product (Ambroxide **14**) and its application. Moreover, we have specified the *E,E*-isomer that was used for the biotransformations (*E,E*-homofarnesol **13** and *E,E*-homofarnesoic acid **17**).

Page 12, line 359, preparation of spheroplasts. The preparation of spheroplasts requires sucrose, NaCl, EDTA, lysozyme. Page 60, line 60, legend to Figure 1. "using detergents enhances mass transfer but increases costs and creates additional waste. Looking at these two points, the question is still open what is best in terms of cost and waste production. In that view the author did not address the point raised earlier. The production of each of these ingredients has a cost and produces waste. The question remains open if it is less costly and waste generating to (i) produce the cells and engage them in a bioconversion where is only added some detergent, or (ii) produce the cells and run them through the preparation of spheroplasts requiring four ingredients prior to running the bioconversion. At what cost (price and waste is produced e.g. lysozyme, EDTA)? This is not necessarily clear. The point here is not to claim the authors are right or wrong in their conclusion regarding cost and waste. It is simply not possible to conclude on such a topic without having carried out a complete analysis regarding waste and cost of the 2 processes. This is why the authors should refrain of making such not well-founded but "cheap" statements. Having not carried out this analysis for making conclusions based on facts, it should be left at one's own discretion what is better rather than making subjective conclusions.

We agree with the reviewer, further studies are needed to determine if the statement is accurately right. Hence, we have rephrased the sentence in the caption of Figure 1 making our statement less categorical and pointing out the need for more detailed studies on this regard (specially on larger scale). Nonetheless, we believe spheroplasts, and specially CLS, contribute to reduce the costs and waste production of the biotransformations since no additives are needed in the reaction and CLS allow for

the reuse of the biocatalyst. To support our statement, we have performed a preliminary assessment of costs that is included in the ESI as Table S10.

Page 6 line 187. Spheroplast preparation surpassed the reported optimal setup of whole cells treated with SDS. This is the author's result in the comparison they did. However, from where comes the judgement that the reported setup is "optimal", as claimed by the authors? It is here probably sufficient to refer to a "reported setup". Here also the authors make a subjective conclusion about what was published elsewhere.

In the reference 10 (Eichhorn, E. *et al.* 2018), an intensive study of the reaction conditions (pH, SDS concentration, cell density, temperature, etc.) for the cyclization of *E,E*-homofarnesol towards Ambroxide was performed. We used the 'optimal conditions' reported before for our biotransformations. Nonetheless, to avoid subjective conclusions we have removed "*the reported optimal setup*".

Figure S9. I assume that with the substrates listed the authors have looked for some of the substrates at the conversion of only one particular isomer? As is indicated in Figure 2. Same care should then be taken here for the annotations of Fig S9 (13, 17?)

E,E-homofarnesol **13** and *E,E*-homofarnesoic acid **17** were used as pure isomers in the biotransformations. This has been specified in the Fig. S9.

Reviewer #2 (Remarks to the Author):

The authors performed new experiments that were clear enough to answer all my previous concerns, and the manuscript was accordingly amended. At least in what regards spheroplast generation and manipulation, I think the work presented is correct.

We are grateful to the reviewer 2 for their comments.

Reviewer #3 (Remarks to the Author):

The authors have improved the manuscript considerably and have taken into account most of the reviewers' suggestions. However, a few points have not yet been satisfactorily elaborated:

1. The response and corresponding changes regarding "the enhanced mass transfer that takes place in continuous flow reactors," which refers to the corresponding author's review paper, are not correct, especially since the reactor type is not specified. For example, a stirred tank reactor can be operated in either batch or continuous mode, and mass transfer is enhanced only by the stirring rate. Similarly, mass transfer in tubular reactors, spinning disk reactors, etc., depends on flow conditions and, for heterogeneous catalysts, also on packing (which determines channeling and pressure drop) and particle size and porosity (which determine internal mass transfer). The authors should consider continuous processing with immobilized biocatalysts as a step toward increasing the productivity of biocatalytic processes (process intensification), which is also related to the miniaturization of flow reactors (more on this topic in doi:10.3390/catal9030262 and doi:10.1016/j.cogsc.2021.100546).

We have replaced our previous statement "the enhanced mass transfer that takes place in continuous flow reactors," by "we considered continuous processing as a step towards increasing the productivity of the SHC biotransformations" as suggested by the reviewer.

2. The use of the term "enzyme" instead of "protein" or "preparation" in the reported concentrations or calculated productivities is not correct. Since the preparations are not purified enzymes, this should be carefully checked and corrected. Biocatalyst productivity in Table 1 is defined per enzyme but should be reported per protein, and biocatalyst concentrations in Figure 2 are defined as mg of enzyme per mL but should be reported in mg of protein, better yet, in U/mL. In the supporting text to Figure 4, the "enzyme" concentration is given, which should better be changed to "enzyme preparation", which is correctly used p.23 of SI to denote the partly purified preparation.

In agreement with the reviewer, we have replaced the term "enzyme" by "protein" in Table 1, Figure 3 and the supporting text to Figure S4.

3. The problem with the legend in Figure 3C referred to only 2 results and 4 subjects in the legend. It was good to omit the two cases with zero values and provide this information in the caption. However, titles of Y-axes in Figures 3B and 3C should be consistent with the other graphs/tables and use "Molar conversion" and not the abbreviation m.c.

The abbreviations "m.c." have been replaced by "Molar conversion" in Figure 4b and c.

4. The authors have included more data and images of CLSs, which is welcome. However, Figure S112B is not very informative in its current state (the CLSs are almost not visible), and the macroscopic image of Figure S112C should also be enlarged to focus on the particles and less on the ruler to give a clearer picture of the CLSs.

We have enlarged the previous image in Figure S12C and we have also added another image of the CLS in solution to complement the Figure S12B. In Figure S12B, we meant to show the visual differences between the CLS (either with glutaraldehyde or polyethyleneimine) and after the treatment

with BDE which did not work and thus it has a similar look to the spheroplasts. In this way, one can easily recognize if the crosslinking protocol is successful. In addition, we have replaced the image in Figure 3A with a more informative image where spheroplasts and CLS in suspension can be observed.

5. It was good to compare immobilization yields of various immobilization techniques tested. Can you also compare the retained activities of obtained preparations?

We have added a new column with the retained activities to Table S9. However, we would like to point out that the retained activities were calculated comparing the conversion (%) of the immobilized and free biocatalyst preparations after 24h-biotransformations.

REVIEWER COMMENTS

Reviewer #1 (Remarks to the Author):

The authors addressed my concerns last raised in the second review round, including making less categorical statements when no data is available for firmly comparing processes regarding cost and waste generation. This might be the content of a study on its own.

I would like to thank the authors for sharing this exciting piece of work, which gives new insights on the use of spheroplasts in biocatalytic reactions in general.

I have the two following minor points:

I am not sure I could find Table S10 in the ESI file I could download today, and in which the authors made a preliminary assessment of costs/waste generation? Is Table S10 really missing? No table found having the content announced for Table S10.

Title of legend to Figure 3 (page 8 line 194): should probably read Squalene-hopene-cyclase-catalyzed cyclization of ... ("cyclase" is missing).

Reviewer #3 (Remarks to the Author):

The authors have adequately addressed all the comments/suggestions of the reviewer. The manuscript can be accepted for publication.

REVIEWER COMMENTS

(Answers by authors in blue)

Spheroplasts preparation boosts the catalytic potential of a terpene cyclase

Ana I. Benítez-Mateos, Andreas Schneider, Eimear Hegarty, Bernhard Hauer*, and Francesca Paradisi*

Reviewer #1 (Remarks to the Author):

The authors addressed my concerns last raised in the second review round, including making less categorical statements when no data is available for firmly comparing processes regarding cost and waste generation. This might be the content of a study on its own. I would like to thank the authors for sharing this exciting piece of work, which gives new insights on the use of spheroplasts in biocatalytic reactions in general. I have the two following minor points:

I am not sure I could find Table S10 in the ESI file I could download today, and in which the authors made a preliminary assessment of costs/waste generation? Is Table S10 really missing? No table found having the content announced for Table S10.

We apologize for the missing data, Table S10 is now included in the ESI (page 18).

Title of legend to Figure 3 (page 8 line 194): should probably read Squalene-hopene-cyclase-catalyzed cyclization of ... ("cyclase" is missing).

The title of legend to Figure 3 has been amended as suggested by the reviewer.

Reviewer #3 (Remarks to the Author):

The authors have adequately addressed all the comments/suggestions of the reviewer. The manuscript can be accepted for publication.

We thank the reviewer for the comments.

REVIEWERS' COMMENTS

Reviewer #1 (Remarks to the Author):

The authors addressed the two points I mentioned in the last review cycle.